# Large-scale photonic network with squeezed vacuum states for molecular vibronic spectroscopy

Hui Hui Zhu[1,10], Hao Sen Chen[2,10], Tian Chen [2] ✉, Yuan Li[1], Shao Bo Luo[3], Muhammad Faeyz Karim[1], Xian Shu Luo[4], Feng Gao[4], Qiang Li[4], Hong Cai[5], Lip Ket Chin[6] ✉, Leong Chuan Kwek[1,7] ✉, Bengt Nordén[8] ✉, Xiang Dong Zhang [2] ✉ & Ai Qun Liu [1,9] ✉

Although molecular vibronic spectra generation is pivotal for chemical analysis, tackling such exponentially complex tasks on classical computers remains inefficient. Quantum simulation, though theoretically promising, faces technological challenges in experimentally extracting vibronic spectra for molecules with multiple modes. Here, we propose a nontrivial algorithm to generate the vibronic spectra using states with zero displacements (squeezed vacuum states) coupled to a linear optical network, offering ease of experimental implementation. We also fabricate an integrated quantum photonic microprocessor chip as a versatile simulation platform containing 16 modes of single-mode squeezed vacuum states and a fully programmable interferometer network. Molecular vibronic spectra of formic acid and thymine under the Condon approximation are simulated using the quantum microprocessor chip with high reconstructed fidelity ( > 92%). Furthermore, vibronic spectra of naphthalene, phenanthrene, and benzene under the non-Condon approximation are also experimentally simulated. Such demonstrations could pave the way for solving complicated quantum chemistry problems involving vibronic spectra and computational tasks beyond the reach of classical computers.

Exploration of molecular vibronic spectra is vital for understanding molecular properties in chemical analysis and biological labels[1–7]. Vibronic spectroscopy typically involves characterizing molecular vibronic transitions, which are simultaneously changed in a molecule's electronic and vibrational energy levels[8]. The resulting vibronic transition probabilities between two electronic states are proportional to the Franck-Condon factors (FCF)[8–10]. Given its pivotal role in chemistry, various strategies on classical computers have been extensively studied. Among them, one renowned algorithm is the eigenvalue-trace algorithm, in which the running time scales

[1]Quantum Science and Engineering Centre (QSec), Nanyang Technological University, Singapore, Singapore. [2]Key Laboratory of Advanced Optoelectronic Quantum Architecture and Measurements of Ministry of Education, School of Physics, Beijing Institute of Technology, Beijing, China. [3]School of Microelectronics, Southern University of Science and Technology, Shenzhen, China. [4]Advanced Micro Foundry, Singapore, Singapore. [5]Institute of Microelectronics, A*STAR (Agency for Science, Technology, and Research), Singapore, Singapore. [6]Department of Electrical Engineering, City University of Hong Kong, Hong Kong SAR, China. [7]Centre for Quantum Technologies, National University of Singapore, Singapore, Singapore. [8]Department of Chemistry and Chemical Engineering, Chalmers University of Technology, Gothenburg, Sweden. [9]Institute of Quantum Technology (IQT), The Hong Kong Polytechnic University, Hong Kong SAR, China. [10]These authors contributed equally: Hui Hui Zhu, Hao Sen Chen. ✉e-mail: chentian@bit.edu.cn; lkchin@cityu.edu.hk; cqtklc@nus.edu.sg; norden@chalmers.se; zhangxd@bit.edu.cn; eaqliu@ntu.edu.sg

exponentially in the system size[11,12]. A quantum-inspired classical algorithm is also proposed to obtain the partial solution of the molecular vibronic spectra simulation[13]. Thus, it remains a long-standing computationally difficult problem that cannot be efficiently solved using classical computers. Recently, the advent of quantum simulation[14–16], a groundbreaking development at the intersection of computer science and quantum mechanics, holds the promise of overcoming the computational challenges associated with exponential computing time, opening up new avenues for advanced molecular studies.

Quantum algorithms for the simulation of molecular vibronic spectra have been demonstrated in systems with superconducting qubits[17] and trapped ions[18,19], such as the experimental implementation of a boson sampling protocol and construction of the vibrational Hamiltonian in a standard quantum circuit. Yet, the study of molecules with multiple modes is hampered by the limited gate fidelities or the susceptibility to various noise sources in superconducting or trapped-ion systems[20,21]. In addition, the low working temperature requirements in the two systems also pose challenges for system miniaturization[22,23]. As a result, only molecules with two vibronic modes have been achieved experimentally. Meanwhile, the advantages of photons, such as their ease of manipulation, precise modulation, and capacity to operate at room temperature, underscore their potential for photonic systems as a promising platform for molecular simulation[24]. Notably, for the simulation of vibronic spectra, the vibrational transitions between two electronic states of a molecule can be mapped into the photon transitions in linear optical quantum systems. By employing the boson sampling protocol with prepared squeezed coherent states, analogies between vibrational modes in molecules and optical modes in waveguides may facilitate this simulation efficiently[10,24].

However, in integrated and bulky optical systems, only the vibronic spectra of virtual molecules or actual two-mode molecules have been experimentally realized[24,25]. In bulky optical systems, the practical implementation of a modulated coherent state and squeezed vacuum state entails bulky components, such as plates, lenses, and modulators, for phase locking, hindering the devices' scalability[26–28]. For integrated photonic microprocessor chips, the constraints on the

squeezing level and the broadband spectral characteristic of the squeezed source result in limited interference visibility between the coherent and the squeezed light[24,26]. Consequently, the on-chip realization of vibronic simulations of actual molecules with multiple modes using squeezed coherent states remains an ongoing challenge.

Here, we propose a nontrivial method and demonstrate an integrated quantum photonic chip for the molecular simulation. The scheme injects light with squeezed vacuum states, instead of conventional squeezed coherent ones, to be injected into a linear optical network to calculate FCF easily. The chip comprises sixteen nonlinear optical sources and a generalized reprogrammable interferometer network to implement squeezing, rotation, and photon-number-resolving operations experimentally. We use the integrated quantum photonic chip for the proof-of-concept demonstrations of simulating the vibronic spectra of formic acid ($CH_2O_2$) with four selected modes and thymine ($C_5H_6N_2O_2$) with seven selected modes under the Condon approximation. Meanwhile, the vibronic transitions of naphthalene, phenanthrene, and benzene are experimentally observed under the non-Condon approximation. Our scheme and integrated quantum photonic chip open new avenues for numerous practical applications in quantum chemistry, such as molecular docking problems[29] and quantum machine learning, including graph classification[30].

## Results

### A squeezed vacuum state and a linear optical network for molecular vibronic spectra approximation

Molecular vibronic spectra describe the molecular vibronic transition, and the vibronic spectral profile, called the Franck-Condon profile (FCP), which is obtained by computing the corresponding FCF, i.e., the square of the overlap integral of two wave functions in different electronic levels within the Born-Oppenheimer approximation and the Condon approximation[31–33]. In classical computing, solving the classically challenging problem of computing the FCF for a given vibronic molecular transition is equivalent to the loop hafnian of a particular matrix, which is a quantity related to the perfect matchings of a graph with loops[6,13,31]. The development of quantum computing puts forward a new way, which leads to an efficient solution to the complicated calculation of FCF in the quantum framework. For a molecule with $n$

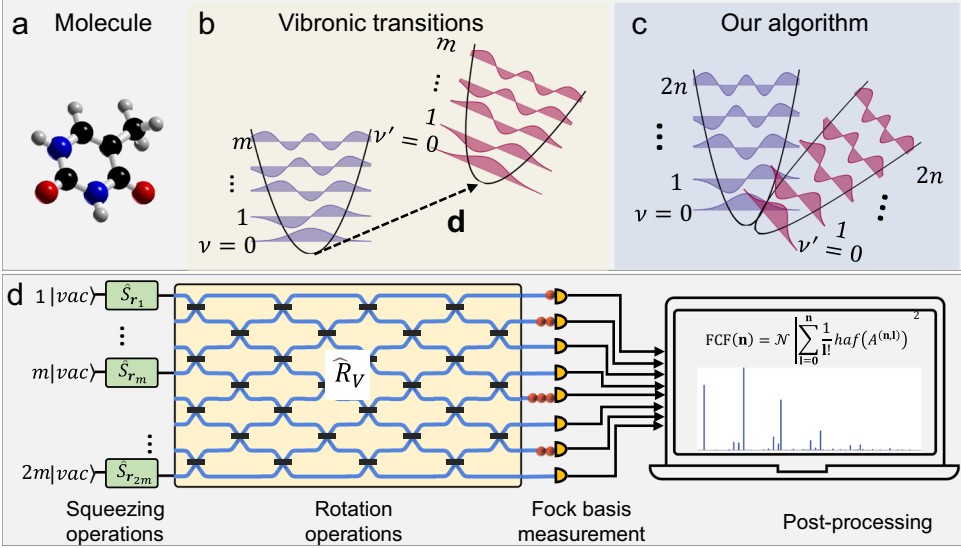

**Fig. 1 | A squeezed vacuum state and a linear photonic network for molecular vibronic spectroscopy. a** Molecular structure. **b** A vibronic transition of a molecule includes displacement, squeezing and rotation operations. **c** In our algorithm, by extending $m$ vibronic modes to $2m$ optical modes and applying straightforward rotation and squeezing operations, it is sufficient to reproduce vibronic spectra.

**d** Photonic quantum circuit model, by translating the Doktorov operator to squeezing operators $\hat{S}(r)$ and rotation operator $\hat{R}_V$. Output photon sampling distribution followed by post-processing can generate the vibronic spectroscopy of the molecule.

vibrational modes, as shown in Fig. 1a, the quantum computation of the vibronic spectra needs to consider the transformation between two potential energy surfaces. The normal coordinates of the final and initial electronic states are linearly related by the Duschinsky transformation[34], which can be represented by the Doktorov operator $\hat{U}_{Dok}$ and further expressed in three quantum operators: squeezing, rotation, and displacement (see detailed in Supplementary Note 1). As shown in Fig. 1b, the previous quantum computation for FCF requires producing squeezed coherent states with $m$ modes. However, the experimental obstacle in squeezed coherent state preparation[27] limits the application of this method.

To perform the quantum computation of FCF efficiently in the experiment, we build a relation between the loop hafnian and the hafnian (the number of perfect matchings of a graph without loops). Since the boson sampling protocol establishes a link between the hafnian matrix function and the output pattern within easily attainable linear optics[35–37], it is straightforward to implement the FCF experimentally within the quantum frame. Through the mode expansion, the original displacement parameter of the $i$-th vibrational mode is represented by the covariance between the $i$-th and $(m+i)$-th optical modes (see detailed derivation in Supplementary Note 1). The original vibrational transition with $m$ modes is related to squeezing and rotation operations with $2m$ optical modes, as shown in Fig. 1c. The established relation between FCF and the hafnian function of matrix $B$ and its extension, matrix $A$, is

$$
\begin{aligned}
FCF(\mathbf{n}) &= \mathcal{N}\left| lhaf\left(\bar{B}^{(\mathbf{n})}\right)\right|^2 \\
&= \mathcal{N}\left|\sum_{\mathbf{l}=\mathbf{0}}^{\mathbf{n}} \frac{1}{\mathbf{l}!} haf\left(A^{(\mathbf{n},\mathbf{l})}\right)\right|^2,
\end{aligned}
\tag{1}
$$

where $\bar{B}$ is a matrix generated by replacing the diagonal entries of $B$ with displacement vector $\mathbf{K}$, in which $B$ is the matrix correlated to the molecular vibronic parameters (detailed in Supplementary Note 1). $\mathcal{N}$ is a normalization constant, $lhaf()$ is the loop hafnian function of the matrix, $haf(\cdot)$ is the hafnian function of the matrix, and $(\mathbf{n},\mathbf{l})$ is an output pattern. We denote $(\mathbf{n},\mathbf{l})$ by $(n_1,n_2,\ldots,n_m,l_1,l_2,\ldots,l_m)$, where $n_k$ corresponds to the number of photons in the $k$ th mode and $l_k$ corresponds to the number of photons in the $(m+k)$ th mode. $A^{(\mathbf{n},\mathbf{l})}$ is a submatrix of $A$ depending on the measured output pattern $(\mathbf{n},\mathbf{l})$ with the matrix $A = \begin{bmatrix} B & Z \\ Z & 0 \end{bmatrix}$, in which $Z$ is a diagonal matrix associated with the displacement vector between the equilibrium positions of the energy surfaces.

Meanwhile, the probability amplitudes of an output pattern are proportional to the square of the hafnian function, $Pr(\mathbf{n},\mathbf{l}) = \frac{1}{\mathbf{n}!\mathbf{l}!\sqrt{\sigma_Q}}\left|haf(A^{(\mathbf{n},\mathbf{l})})\right|^2$, where $\sigma_Q = \sigma + I_{4m}/2$ with $I_{4m}$ being the $4m \times 4m$ identity matrix and $\sigma$ being the covariance matrix[37,38]. Moreover, a sign approximation is used such that the sign of the hafnian in the summation of Eq. (1) is ignored (see detailed discussion in Supplementary Note 1). Therefore, the relationship between $Pr(\mathbf{n},\mathbf{l})$ and the FCF can be established, where the sampling results are used to approximate the molecular vibronic spectra. The approximated FCF ($\widetilde{FCF}(\mathbf{n})$) at $0K$ is expressed as

$$
\begin{aligned}
\widetilde{FCF}(\mathbf{n}) &= \mathcal{N}\left|\sum_{\mathbf{l}=\mathbf{0}}^{\mathbf{n}} \frac{1}{\mathbf{l}!}\left|haf\left(A^{(\mathbf{n},\mathbf{l})}\right)\right|\right|^2 \\
&= \mathcal{N}'\left|\sum_{\mathbf{l}=\mathbf{0}}^{\mathbf{n}} \left(\frac{Pr(\mathbf{n},\mathbf{l})}{\mathbf{l}!}\right)^{\frac{1}{2}}\right|^2,
\end{aligned}
\tag{2}
$$

where $\mathcal{N}'$ is a normalization constant and $Pr(\mathbf{n},\mathbf{l})$ is the probability of measuring an output pattern $(\mathbf{n},\mathbf{l})$. Subsequently, by sampling many FCFs, the approximated FCP at each given vibrational transition

frequency ($\omega_{vib}$) is obtained as

$$
\widetilde{FCP}(\omega_{vib}) = \sum_{\mathbf{n}}^{\infty} \widetilde{FCF}(\mathbf{n})\delta\left(\omega_{vib} - \sum_k \omega'_k n_k\right),
\tag{3}
$$

where $\{\omega'_k\}$ are the harmonic angular frequencies of the final and initial states and $n_k$ corresponds to the number of photons in the $k$ th mode. Combined with Eq. (2), our algorithm can estimate the FCP by stochastically sampling the known probability distribution for the output modes, and its scaling behavior is described in "Methods" section and Supplementary Note 2. Our algorithm is a straightforward approach that comprises four components, as depicted in Fig. 1d: squeezing operations $\hat{S}_r$, rotation operations $\hat{R}_V$, Fock basis measurement and post-processing, which can be accomplished without needing a displacement term. The squeezing value $\mathbf{r}$ and unitary matrix $V$ are given by the Takagi-Autonne decomposition[39] of matrix $A$, resulting in $A = V^T(\tanh(\mathbf{r}))V$. After the Fock basis measurement[40], the molecular vibronic spectra are obtained by post-processing the output photon sampling distribution.

## Quantum experimental framework

The correlation between FCF and the hafnian function allows us to implement the quantum optics simulation of molecular vibronic spectroscopy. An overview diagram of the entire system is shown in Fig. 2, which consists of three subsystems: (1) dual pump preparation, (2) quantum microprocessor, and (3) control and measurement. First, the pump light is prepared with integrated filters based on asymmetric Mach-Zehnder interferometers (MZIs) using the dual-pump scheme (see characteristic source analysis in Fig. S2) as shown in Fig. 2a. To increase the input light power for enlarging the squeezing level of the squeezed vacuum state, an erbium-doped fiber amplifier is connected to amplify the laser power, and a pair of wavelength division multiplexer (WDM) is used to filter the pump signal. The integrated quantum microprocessor for vibronic transition profile reconstruction is illustrated in Fig. 2b. The squeezing source is applied via a degenerate spontaneous four-wave mixing process from the spiral lines, which is a single-pass squeezer. The unitary matrix is then performed in the evolution through the MZI network. As the programmable gate network embedded in the chip can be reconfigured as large as $16 \times 16$ dimensions, the implemented unitary operation can be tuned to match a particular target molecule. Besides, a pseudo-number-resolving detection scheme (up to 4 photons) is applied to achieve the photon number resolution at the photon counter. This scheme can be achieved by adding the 50:50 on-chip beam splitters and detecting all single outputs with the superconducting single-photon detectors. This approach could probabilistically split them at three beam splitters when four photons exist in a single output mode. Subsequently, the output coincidence photon numbers are also probabilistic. The detailed analysis of the probabilistic photon number-resolving is described in Supplementary Note 3. Finally, as shown in Fig. 2c, a fiber array is used to couple light out of the chip, and the output photons pass through the polarization controllers and the filters. Single-photon detectors finally collect them with a time tagger. The server computer controls the operating conditions for calculating the molecular vibrational spectra, including a thermoelectric controller and water-cooling temperature controller under the chip, the electronic control of squeezing level of the quantum sources and linear optical circuit, and the photon number resolving detection.

The quantum microprocessor chip is $13 \times 4$ mm$^2$, monolithically comprising mainly 16 photon sources, 272 thermo-optic phase shifters, 319 multimode interferometer beam splitters, and 65 optical couplers. The chip is packaged optically, electronically, and thermally, featuring high-density electrodes mounted to a printed circuit board through double-line wire bonding. The details of the experimental setup are discussed in Supplementary Note 3. The reconfigurability and control of the unitary transformation are detailed in

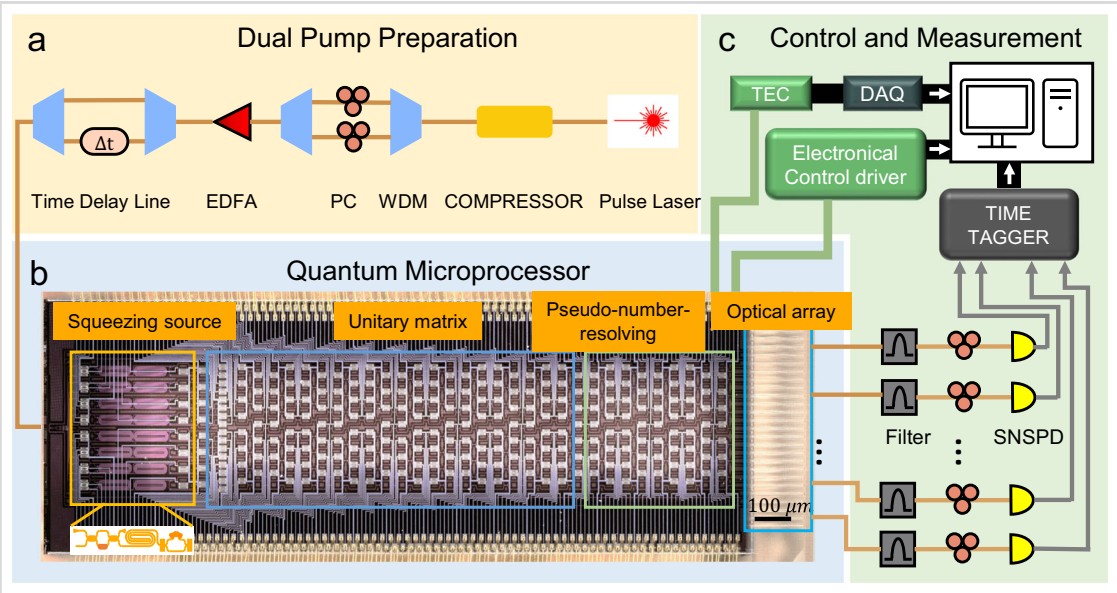

**Fig. 2 | Schematic of a quantum microprocessor chip and experimental setup.**
**a** Setup to generate single-mode squeezing using dual pumps from 2-ps laser pulses, consisting of an optical pulse compressor, an erbium-doped fiber amplifier (EDFA), wavelength-division multiplexers (WDM), and a polarization controller (PC). The two pumps at the wavelength of 1546 nm and 1553 nm are selected and combined using WDM and coupled into the chip through a grating coupler. **b** Photograph of the microprocessor chip. Spiral sources are used to produce a vacuum-squeezed state. Programmable interferometer network is designed to achieve an arbitrary unitary matrix, and on-chip beam splitters are used for pseudo-number-resolving detection. Input single pump light is coupled to the chip by a one-dimensional subwavelength grating coupler. Output photons are coupled by edge couplers to fiber arrays. The scale bar denotes $100 \, \mu m$. **c** An overview of off-chip control and measurement devices, including temperature controller (TEC), electrical modulators, output photon detection (Time Tagger), Analog-to-Digital Converter (DAQ), and data processing with a server computer.

Supplementary Note 4, and the achieved fidelities of the reconstructed matrix are presented in Supplementary Fig. 3. Some basic quantum characterizations of the circuit are shown in Supplementary Fig. 4, which manifests that the quantum photonic chip could be used to study molecular vibronic spectra.

**Molecular vibronic spectra**

The quantum microprocessor chip is employed to compute the FCPs of two molecules: formic acid ($CH_2O_2$) and thymine ($C_5H_6N_2O_2$), which possess a considerable number of vibrational modes, specifically 9 modes for formic acid and 39 modes for thymine. Due to the chip's size limitation, we deliberately exclude specific modes with relatively small FCFs. More details of the chosen vibrational frequencies and local basis transformations in our experiment can be found in Supplementary Note 5. The theoretical FCPs are calculated with the vibronic structure program package *hotFCHT*[32,41]. The experimental encoding and calculation details are listed in Supplementary Note 6.

In the experiment, for formic acid ($CH_2O_2$) with 9 vibrational modes, only 4 quantum harmonic oscillators are selected and then effectively doubled and encoded to an 8-mode network. The incoming photons from the 1st and 5th modes are probabilistically split into four waveguides via three $1 \times 2$ beam splitters. The theoretical and experimental FCPs for formic acid are presented in Fig. 3a as vertical gray bars above and below the x-axis, respectively. The red and blue curves represent the Lorentzian broadening of the bars, depicting the theoretical and experimental spectra, respectively. The similarity between the experimentally reconstructed and theoretical FCPs is characterized by computing the fidelity of two sequences, $F = \sum_i \sqrt{p_i q_i}$, where $p$ and $q$ are the normalized theoretical and experimental FCF sequences of the molecules, respectively[42-44]. A fidelity $F$ of 92.9% is achieved in this case, which is limited by the inevitable flaws in circuit fabrication and operation, photon noise, and photon loss. In particular, the loss affects the FCF values at high frequencies. Because of the photon loss, only a maximum of four-photon coincidence is achievable. Therefore,

events with more than four photons cannot be stored, and the FCF values are further reduced at high frequencies (see a more detailed discussion in Supplementary Note 7). We further calculate the difference between the fidelity of the reconstructed FC profile $F_Q$ and the optimal fidelity obtainable from a classical strategy $F_C(C = F_Q - F_C)$ to benchmark against the classical simulation[25,42]. The result shows an improvement in the fidelity of the experimentally reconstructed FCP to the ideal FCP over the classical strategy ($C = 6.8\%$) for formic acid molecules at the experimental maximum squeezing level (-0.3). The simulated theoretical fidelity difference between quantum and classical methods versus squeezing values under different loss values for formic acid molecules is shown in Supplementary Fig. 8, which shows that $C$ is ~10% for the case without considering any losses (at -0.3 squeezing level).

Calculating the FCP of thymine ($C_5H_6N_2O_2$) presents a more intricate task due to its vibronic boson sampling involving 39 modes. To manage this complexity, we chose to work with a subset of 7 modes, which have been meticulously encoded into our chip. Furthermore, the selected 1st, 6th, and 7th modes are probabilistically split into two waveguides via a $1 \times 2$ beam splitter. Figure 3b depicts the theoretical and experimental spectra, including the line-broadening effects (red and blue curves). The fidelity is 97.4%, corresponding to $C = 7.3\%$. The fidelity is limited to the experimental mode number, leading to the loss of some frequencies upon detection. Moreover, the truncation of the FCP at 4-photon contributions also ignores the values in some frequencies. In addition, photon loss, leading to the misclassification of specific multiphoton events as single photons, inflates the count of single-photon events. A detailed discussion of these errors is found in Supplementary Note 7. Nonetheless, it can reconstruct most of the peaks in this profile. Details of the selected vibrational modes can be found in Supplementary Note 5. In conclusion, the vibronic spectroscopies of two molecules are simulated experimentally on the quantum microprocessor chip. While the classical computation of FCP is easier by relying on the Condon approximation, it is essential to

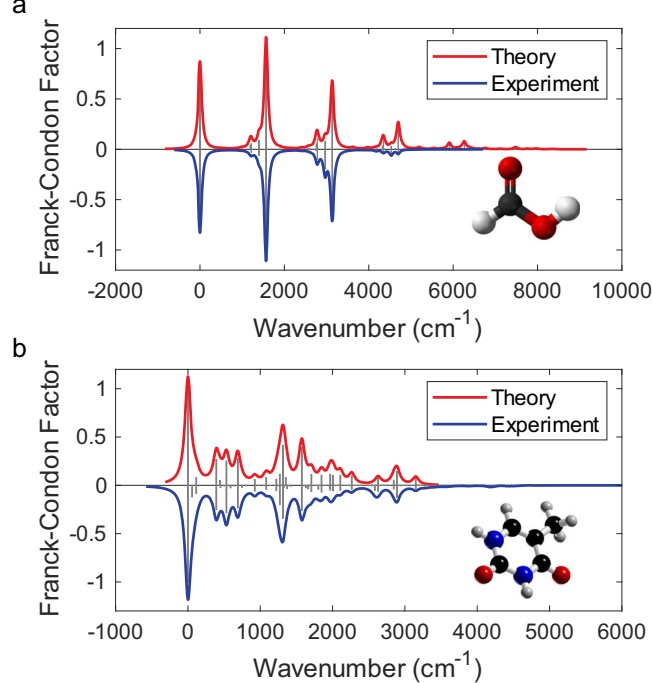

**Fig. 3 | Vibronic spectra reconstruction.** Franck−Condon profiles are obtained from chip distributions programmed according to the vibronic transitions of formic acid (**a**, with the structure shown in the inset) and thymine (**b**, with the structure shown in the inset). Gray bar graphs depict the histogram of energies, whereas red and blue continuous curves show the Lorentzian broadening of the bars.

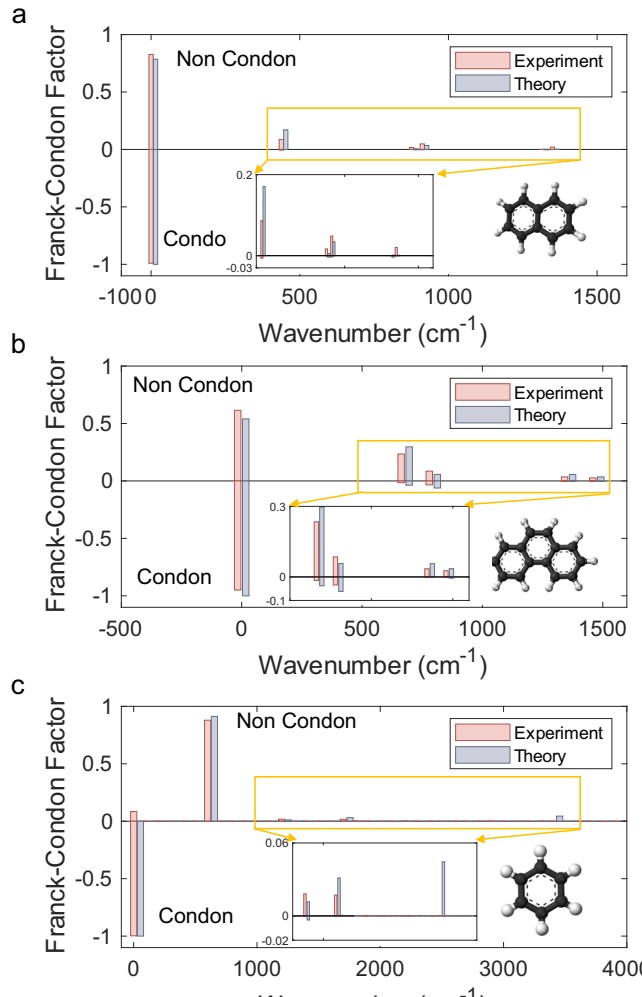

**Fig. 4 | Vibronic spectra with non-Condon effects.** Non-Condon and Franck −Condon profiles are obtained from chip distributions programmed according to the vibronic transitions of naphthalene (**a**, with structure shown in the inset), phenanthrene (**b**, with the structure shown in the inset), and benzene (**c**, with the structure shown in the inset). Red bar graphs depict the histogram of experimental energies, whereas blue bars show the theoretical results. Insets are enlarged parts of small peaks.

acknowledge that, for many large systems, FC-forbidden or weakly FC-allowed transitions cannot be excluded, and the FC assumption is known to break down in some cases (most notably in aromatic molecules). In such instances, it becomes imperative to introduce corrections to ensure a meaningful interpretation of the observed transitions.

## Non-Condon effects in molecular spectroscopy

The Condon approximation, which assumes a constant transition dipole moment, may no longer suffice in simulating some molecules[45]. To improve the accuracy, it is necessary to include the non-Condon approach. Reference[46] shows that a non-Condon approximation could be considered and implemented on linear optical quantum hardware. A non-Condon profile is approximated with Gaussian states through the first- and second-order Herzberg−Teller expansions of the transition dipole moment operator. Then, following our previous derivation, all these Gaussian states can be implemented on the integrated quantum photonic chip. The derivation details can be found in Supplementary Note 8.

As a proof of principle, the vibronic spectra of naphthalene ($C_{10}H_8$), phenanthrene ($C_{14}H_{10}$), and benzene ($C_6H_6$) for the first-order Herzberg−Teller expansion are experimentally demonstrated in Fig. 4. The measured fidelities with the vibronic spectra of naphthalene, phenanthrene, and benzene are 98.4%, 98.4%, and 93.4%, respectively. To illustrate the non-Condon approximation, the computed FCPs are compared to non-Condon scenarios by omitting the transition dipole moment operator and exclusively considering the Doktorov operators. The molecular characteristic parameters, including vibrational frequencies and transformation matrices, are summarized in Supplementary Note 5. The $\tau$ resulting from the quadratic truncation error $O(\tau^2)$ for approximating the nonunitary operator as a linear combination of unitary operators in the experiment is set to 1. Though this signifies that a smaller $\tau$ leads to a more minor error, the smallest

possible $\tau$ is dictated by the sampling error bars in our integrated chip device. For naphthalene and phenanthrene, only the two vibrational modes of the molecules that are relevant to demonstrate our method are considered, and photons from the two modes are probabilistically split into four channels. As shown in Fig. 4a, naphthalene does not exhibit vibronic spectral progression in the Condon regime due to the absence of displacement in its vibronic transition. In contrast, when considering the linear HT operator, the vibronic spectral progression is obtained (see red bars located above the x-axis in Fig. 4a). Non-negligible non-Condon effects in the FCP of phenanthrene are also observed, resulting in neighboring peaks at higher wavenumbers due to the Duschinsky mode mixing effect (Fig. 4b). The experimental results (red bars) match well with the theoretical ones (blue bars).

After the obtained simulation for naphthalene and phenanthrene by restricting to two modes, it is crucial to evaluate the precision of our experimental platform in a broader scenario. Figure 4c shows that the experimental vibronic spectrum of benzene corresponds to the $e_{2g}$ symmetry block for the first-order HT expansion (detailed parameter information in Supplementary Note 5 and Ref[46]). The experiment considers five vibrational modes, with the number of photons per mode

 

limited to two. Figure 4c shows that the generated profile from our experiment (red bars above the $x$-axis) matches the profile computed by the classical algorithm (blue bars). In addition, the non-Condon spectrum with the FCP (red bars below the $x$-axis) reveals a discernible contrast. As a result, integrating the new algorithm with the linear optical quantum simulation platform enables the experimental simulation of intricate molecular processes beyond the Condon regime, thereby broadening the range of potential applications for boson sampling.

## Discussion

This work proposes a nontrivial theoretical model employing a linear optical network circuit and squeezed vacuum sources to simulate molecular vibronic spectra. Modifying from squeezed coherent to squeezed vacuum simplifies the experimental implementation for molecular simulation, enabling the simulation of complex molecules with multiple modes on integrated photonics circuits. Our approach allows the generation of molecular vibronic spectra through the modified boson sampling in an optical network. Compared with the traditional boson sampling, our algorithm incorporates an additional weighted summation of the hafnian function. It is noted that the number of summation terms is smaller than the sampling number. As a result, the algorithm's scalability mainly depends on the sampling process (see details in "Methods" section and Supplementary Note 2). We want to highlight that some classical algorithms can also achieve the same accuracy as compared to quantum approaches in solving the molecular vibronic spectra, such as Gurvits's algorithm[13,47,48], but are limited to specific cases, such as the Fock-state or Gaussian boson sampling with large squeezing. It remains an interesting question whether one can find a classical algorithm to provide a generalized solution in which quantum approaches may offer the solution with potential quantum advantages[13].

To perform molecular simulation within the harmonic model, we demonstrate an on-chip reconfigurable quantum simulator achieved through an integrated quantum photonic microprocessor chip with 16 squeezed modes injected into an interferometer network. The full programmability of our chip, which conducts arbitrary unitary operations of the interferometer network and the pseudo-number-resolving detection scheme, enables its practical applications in various molecular simulations of complex molecules. Combining the theoretical model with the experimental hardware platform, our chip can simulate the FCP of larger molecules with 7 modes. This surpasses previous research endeavors, which were limited to simulating vibronic spectra exclusively for virtual molecules[24,42] or molecules featuring only two modes, such as tropolone[25], $SO_2$[18,19], $H_2O$[17], and $O_3$[17]. In addition, in molecular simulation, the quantum photonic chip for calculating FCFs is a valuable tool for exploring molecular vibronic spectra and simulating molecular vibrational excitations during vibronic transitions[49]. A detailed discussion is added in Supplementary Note 9.

For molecular vibronic spectra simulation on the quantum photonic chip, two molecules, formic acid and thymine, are first tested under the Condon approximation. Then, vibronic transitions of naphthalene, phenanthrene, and benzene under the non-Condon region are calculated. The reconstructed fidelities of all molecules are higher than 92% with a quantum enhancement, which is limited by the squeezing level, chip losses, and detector efficiency. The low losses and high squeezing level in the integrated architecture could further improve quantum enhancements through waveguide design optimization, increasing etch levels, and using low-loss materials with higher nonlinearity[50]. We envision that our approach could yield an early class of practical molecular simulations that operate beyond classical limits and are promising for achieving quantum speed-ups in relevant biochemical applications.

## Methods
### Scalability analysis
The average number of required samples to estimate FCP($\omega$) up to a desired precision $\epsilon$ scales as $O(\epsilon^{-2})$, which represents a polynomial

overhead in the protocol run-time[10]. Here, we explicitly analyze the computing time cost of resolving FCPs with our algorithm and experimental setup.

The algorithm proceeds as follows. Matrix $A$ is first prepared and encoded into the quantum microprocessor chip. Then, the output modes are measured, and the photon numbers sequence $(\mathbf{n},\mathbf{l})$ is obtained. Subsequently, the frequency bin of the FCP corresponds to the measurement values of $\mathbf{n}$ is located, and its value is increased by one. The sampling is repeated until the estimated statistical error of the average values in each discrete bin of FCP is below the desired precision threshold $\epsilon$. The total number of samples taken is denoted as $N_{smp}$. Finally, the summation is performed to obtain the amplitude of each vibrational frequency.

To access the algorithm, it is rewritten as a stochastic sampling problem, considering a probability distribution determined by the boson sampling device. $\mathbf{X}(\mathbf{n},\mathbf{l})$ is denoted as the frequency of observing the photon number sequence $(\mathbf{n},\mathbf{l})$ in $N_{smp}$ samples, which follows the binomial distribution $\mathbf{X}(\mathbf{n},\mathbf{l}) \sim \mathbf{B}(N_{smp}, \Pr(\mathbf{n},\mathbf{l}))$. In the experiment, the frequency is used to estimate the probability, $\Pr(\mathbf{n},\mathbf{l}) \approx \frac{\mathbf{X}(\mathbf{n},\mathbf{l})}{N_{smp}}$. Since the events that involve different output photons from each port are mutually exclusive, there is a negative correlation between these events. Based on the probability theory[51], the variance of statistical estimation of $\widetilde{FCF}(\mathbf{n})$ is smaller than the sum of the variance of each term,

$$var\left(\widetilde{FCF}(\mathbf{n})\right) < \mathcal{N}\sum_{\mathbf{l}=0}^{\mathbf{n}} \frac{var\left(\sqrt{\frac{\mathbf{X}(\mathbf{n},\mathbf{l})}{N_{smp}}}\right)}{\mathbf{l}!}. \tag{4}$$

Based on the central limit theorem and delta method theorem[50],

$$var\left(\sqrt{\frac{\mathbf{X}(\mathbf{n},\mathbf{l})}{N_{smp}}}\right) \approx \frac{1 - Pr(\mathbf{n},\mathbf{l})}{4N_{smp}} < \frac{1}{4N_{smp}}. \tag{5}$$

By combining Eqs. (4) and (5) and considering only those events that happened,

$$var\left(\widetilde{FCF}(\mathbf{n})\right) < \mathcal{N}\sum_{\mathbf{l}=0}^{\mathbf{n}} \frac{var\left(\sqrt{\frac{\mathbf{X}(\mathbf{n},\mathbf{l})}{N_{smp}}}\right)}{\mathbf{l}!} < \mathcal{N}\frac{t}{4N_{smp}}, \tag{6}$$

where $t$ is the number of summation terms.

If $var\left(\widetilde{FCF}(\mathbf{n})\right) < \epsilon^2$ is required, the number of samples $N_{smp}$ scales as $O(t\epsilon^{-2})$. Meanwhile, $t$ does not increase with the number of vibrational modes in the numerical simulations (see details in Supplementary Note 2). Therefore, the number of expected samples required to achieve convergence of FCP is restricted by a constant determined by the desired precision.

### Experimental setup
The pump laser is generated from an ultrafast optical clock device (PriTel) with a repetition rate of 500 MHz, a central wavelength of 1550.116 nm, and a bandwidth of 1.5 nm. For the dual-pumping scheme, the pulse laser first goes through a compressor (PriTel) to expand the bandwidth to ~10 nm. Then, the two pump wavelengths (1553.33 nm and 1546.92 nm) are selected with a 100 G dense WDM device and recombined into a single-mode fiber. Next, an erbium-doped fiber amplifier is connected to amplify the laser power, and another pair of WDM filters the pump signal. To balance the optical path difference between the two pump wavelengths, tunable delay lines are connected to one arm of the channel to overlap the two pulses. The dual pump light is launched into the device through a subwavelength grating coupler, and photons emerging from the device are then collected through a high-NA fiber array, which has 127 μm spacing and 20 channels. Sixteen off-chip filters (1.2 nm bandwidth, 0.75 dB average insertion loss) are used to remove spurious pump photons and

 

enhance the photon indistinguishability. Photons are then detected by the 16 channels of fiber-coupled superconducting single-photon detectors (Photon SoptTM, 100 Hz dark counts, 85% efficiency). The polarization controllers are added to optimize the polarizations of input/output photons. Finally, the detected photon signal is converted to an electrical signal and processed by the time tagger (Swabian Instrument TM GmbH). Phase shifters on the device are configured through a digital-to-analog converter (q-control) and controlled by the computer. A Peltier controlled by Thorlabs TED200C and a water-cooling system are used to keep the chip temperature constant and reduce the heat crosstalk within the chip.

## Fabrication and packaging

The chip is fabricated using the silicon-on-insulator platform with a 220-nm-thick silicon top layer and a 2-μm thick buried oxide. Subsequently, a thin layer of titanium nitride microheater is deposited in one of the MZI arms based on the thermo-optic effect. To decrease the power consumption further, deep trenches with undercut structures are designed around the thermo-optic phase shifters, and the average power of each MZI is 3.1 mW. The typical waveguide loss analysis is discussed in Supplementary Note 4. For the optical packaging, UV-curable glue is used to attach the fiber array to the chip by adding an index-matched oil. The ending coupling loss is -1.5 dB/facet. For the electrical packaging, we use high-density (two-layer) wire-bonding technology to connect the electrical pads on the chip to the pads of printed circuit boards. With many thermo-optical phase shifters, the accumulated thermal effect on the chip should be considered. A thermoelectric controller and water cooling as the substrate under the chip can be used to control and stabilize the temperature using a temperature controller. The added cooling system can further reduce the heat fluctuations caused by the ambient temperature and the heat crosstalk within the chip.

## Data availability

The data supporting this study's findings are available from the corresponding authors upon request.

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

## Acknowledgements

This work was supported by the Singapore Ministry of Education (MOE) Tier 3 grant (MOE2017-T3-1-001), the Singapore National Research Foundation (NRF) National Natural Science Foundation of China (NSFC) joint grant (NRF2017NRF-NSFC002-014), and National Natural Science Foundation of China (No. 12234004).

## Author contributions

H.H.Z., H.S.C., T.C., X.D.Z., and A.Q.L. jointly conceived the idea. H.S.C., H.H.Z., and A.Q.L. performed the numerical simulations and theoretical analysis. H.H.Z., S.B.L., M. F. K., X.S.L., F.G., Q.L., and H.C. did the experiments. H.H.Z., Y.L., L.K.C., B. N., X.D.Z., L.C.K., and A.Q.L. were involved in the discussion and data analysis. H.H.Z., H.S.C., X.D.Z., L.K.C., B. N., L.C.K., and A.Q.L. prepared the manuscript. X.D.Z. and A.Q.L. supervised and coordinated all the work. All authors commented on the manuscript.

## Competing interests
The authors declare no competing interests.
