## [Peer Review File · Nature Communications]

Large-scale photonic network with squeezed vacuum states for molecular vibronic spectroscopyREVIEWER COMMENTS

Reviewer #1 (Remarks to the Author):

Zhu et al. developed a quantum optical analog simulation method for the molecular vibronic spectra. Unlike the existing approach, the method does not need squeezed coherent states and only requires squeezed vacuum states by embedding displacements in the covariance matrix. The idea is simple but innovative, and it will make the experimental implementation plausible, as the authors demonstrated experimentally for large molecular systems.

Nevertheless, I have a big concern about the quality of the manuscript. I often find mistakes and typos in the manuscripts and supplementary materials, and following the equation derivations in the supplementary material is difficult (and I suspect it would contain mistakes). I stopped reviewing the manuscript and could not proceed further. Authors should improve their manuscript significantly for correctness and clarity. Please recheck every detail.

Reviewer #2 (Remarks to the Author):

Zhu et al. outline their approach to simulating Franck-Condon factors of vibronic spectra using a bosonic network. They introduce a method to remove the need for a noise-inducing photonic coherent displacement by doubling the network size, and they demonstrate their approach for several target molecules with and without the Condon approximation.

Despite the impressive experimental results, the authors appear to have several misunderstandings and errors in the underlying theory which hinder the interpretation of the results. For this reason, I do not believe that this manuscript should be published in Nature Communications. Nonetheless, I will offer my feedback for revisions and clarifications necessary if it is to be published here or elsewhere.

Firstly, could the authors state clearly how they obtained the frequencies and Duschinsky rotation matrices used in this paper? These values are crucial inputs for the Franck-Condon model, and the accuracy of spectra is only as good as the underlying accuracy of the frequencies/rotations that define the potential. Notably, the authors incorrectly state that

formic acid has 7 vibrational modes. The reduction to 7 modes is confusing considering the fact that the peaks in the spectrum are captured in a 4-mode model. The authors then correctly state that thymine has 39 vibrational modes, but only provide parameters for a 7-mode model (which only truly includes 6 modes if the leading value of δ is zero). There is no explanation in the manuscript how the authors chose which modes to use in the reduced models, and in the case of thymine there is no indication in the main text that the reduction was made.

Based on the raw data, it appears that the authors binned frequencies in the spectra in intervals of 50 cm^{-1} . It is not clear why binning is done at all. It leads to extra data handling, since the raw experimental data is based on output frequency and photon count. It also leads to a clear misalignment of peak positions, and mischaracterization of overlapping peaks as a single peak. For example, comparing Fig. 3a in this manuscript to Fig. 3 in ref. 10 shows a clear difference despite identical input parameters. The thymine spectrum shows low-frequency peaks that are not possible if the inputs in the supporting information are correct.

The authors state that "Ref. [27] showed that inharmonicities could be considered and implemented on linear optical quantum hardware". This appears to be a misunderstanding, since ref. 27 and the work done in this manuscript are for harmonic models. Rather, the corresponding work includes the non-Condon, Herzberg-Teller couplings. How did the authors obtain the Herzberg-Teller coupling parameters used in this work? Later, the authors state that "Although the classical computation of the FC profile is easier by exploiting molecular symmetry, molecules often are asymmetry, especially in the case of large molecules, in which non-Condon effects should be considered". What is the relation between non-Condon effects and asymmetry?

Finally, the authors lack any discussion of their work in comparison to past results. The stated purpose is to make boson sampling practical for larger systems, but how does it compare to the work cited in this manuscript? Notably, the Franck-Condon model fails for many large systems where vibronic coupling and environmental noise are prevalent (cf. <https://doi.org/10.1039/D3SC02453A>). Although the authors have written the manuscript in

terms of photoelectron spectra (which is not necessarily the only use of Franck-Condon factors), it should be noted that for many molecules this approach would at best be applicable for the photoelectron spectrum of the ground electronic state of the cation. For some molecules (cf. <https://doi.org/10.1039/B500055F>) even the ground cationic state has strong vibronic coupling.

Other minor comments:

- On line 79, "the investigation of bigger molecule sizes is constrained by the weak coupling of next-nearest neighbors among ions and superconducting qubits". Whereas superconducting qubits are typically only coupled to nearest neighbors, trapped ions have global coupling through trap vibrational modes.
- On line 117, "the square of the overlap integral between the relative vibrational wave functions of different potential energy surfaces within the Born-Oppenheimer approximation". This formulation is within the Born-Oppenheimer approximation and the Condon approximation, which is an important distinction considering the part of the manuscript about non-Condon effects.
- Fig. 2 has several abbreviations that aren't explained in the main text or the caption
- Please clarify the sentence on line 145, "the probability amplitudes of squeezed vacuum state and linear network are quadratic to Hafnian function as ..."
- On line 150, "fork basis" should be Fock basis
- On line 228, "two complex molecules with a point-group symmetry". I would hesitate to call either molecule complex, and all molecules have a point group symmetry of at least C₁.
- Why do parameters for pyrrole appear in the supporting information?

Reviewer #3 (Remarks to the Author):

In this paper the author proposes a new algorithm composed of single-mode vacuum squeezed states sent through a linear interferometer and finally detected by a pseudo-resolved photon detector. They claim that using single-mode squeezed vacuum states instead of single-mode displaced squeezed states as proposed in other protocols actually avoid experimental challenges. The use of single-mode squeezed vacuum states in their algorithm requires doubling the number of modes.

The author fabricated a 16 input mode photonic chip allowing the generation and the manipulation through a programmable interferometer of single-mode squeezed vacuum states.

The authors finally demonstrated the performances of this photonic chip by simulating the molecular vibronic spectra of two molecules the formic acid and thymine with a fidelity higher than 0.91 %. The authors also experimentally stimulated the vibronic spectra of naphthalene, phenanthrene and benzene under non-Condon regime.

During the reading of this paper, I did not see relevant flaws that should prohibit the publication.

This paper presents an advance in quantum simulation as it shows simulation of the vibronic spectra of large scale molecules under non condom effect. However I see three main points which should be clarified to consider these results as significantly relevant to the photonic quantum simulation field.

** The authors state the displacement operation as an experimental challenge to emphasize the simplicity of the presented algorithm; however, it seems to me that incrementing the number of single-mode squeezed vacuum with the exactly the same squeezing parameter is more challenging than implementing a linear operation of displacement. The authors could explain more specifically why this algorithm is easier to implement.

** The reconstructed fidelity for vibronic spectra of naphthalene, phenanthrene and benzene under non-Condon regime is not reported in the paper. Is there a specific reason I missed it ?

** Morother, it should be good if the author show a significant benchmark against classical simulation using this criteria : $C = FQ - FC$ the difference between the fidelity of the reconstructed Franck-Condon profile FQ and the optimal fidelity obtainable from a classical strategy FC issued from the following paper the author should cite :

<https://www.nature.com/articles/s41567-019-0567-8>

<https://iopscience.iop.org/article/10.1088/1361-6455/aaf031>

The data and the method seems valid to me, the theory of vibronic spectra analysis is out of the scope of my expertise, and I was able to understand details within the supplementary note 1. The authors have made sufficient effort to provide all the keys to a good understanding of the analysis.

Sufficient details have been provided concerning the experimental setup and the photonic chip which I believe would allow anyone to reproduce the experiment.

** I however have one concern about the source characterization in the supplementary note 4 A. Source characterization. The authors claimed they are able to determine the squeezing value of their source by second-order order correlation measurement. This part needs to be clarified, and authors should explain how they retrieve the squeezing value of the source with this measurement.

This article is clear and the authors clearly demonstrate their project and what they have achieved. The abstract clearly outlines the challenges and it is easy to understand what the main text will discuss. The conclusion succinctly summarizes what has been accomplished and opens up possibilities for future work.

** Regarding the bibliography, I believe it is well-informed. However, I would like to add the following two papers to enhance the references:

Paesani, S., Ding, Y., Santagati, R. et al. Generation and sampling of quantum states of light in a silicon chip. *Nat. Phys.* 15, 925–929 (2019). <https://doi.org/10.1038/s41567-019-0567-8>
<https://www.nature.com/articles/s41567-019-0567-8>

William R Clements et al 2018 *J. Phys. B: At. Mol. Opt. Phys.* 51 245503
DOI 10.1088/1361-6455/aaf031

<https://iopscience.iop.org/article/10.1088/1361-6455/aaf031>

** I have a few reservations about the lack of precision regarding certain terms related to quantum mechanics or optics in general. For example, terms like "squeezed photons" can not be used for referring to a squeezed vacuum state.

In summary, regarding the clarity and promising result of this article, it could be accepted

for publication, provided that the following major modifications listed with stars (**) in this report are addressed.

Proposed corrections:

Abstract :

47 - "Universal programmable 16-mode interferometer." The universality of an interferometer is questionable

Introduction :

150 - "Fork" → Fock

150 - " $\hat{R}(\tilde{U})$ " → I would prefer the notation in the supplementary note for the rotation operator : \hat{R}_U (RU)

151 - Notation : We would prefer using r as a squeezing parameter and not λ

Quantum experimental framework :

166 - "Squeezing level of photons" gives the impression that the state is a squeezed Fock state. We would rather use "the squeezing parameter" or just the "Squeezing level of the squeezed vacuum state".

166 - Erbium dope fiber amplifier (EDFA) -> Erbium doped fiber amplifier (EDFA)

Overall, this sentence is a little difficult to understand :

'To increase the input light power for enlarging the squeezing level of photons, an Erbium dope fiber amplifier (EDFA) and a pair of Wavelength Division Multiplexer (WDM) are employed to amplify input light and suppress light power at wavelengths.'

Would it be possible to rephrase it ?

168 - Rephrase the whole sentence to be clearer. "To amplify the light and select only the pump wavelength" for example.

170 - The spiral lines are actually not visible on the figure.

Would it be possible to change the resolution of the picture ? or place an inset with a zoom on the squeezer.

170 - Moreover, what is the role of the spiral in the vacuum squeezed state generation ?

Is it a single pass squeezer or a cavity ring ? Could the author be more specific ?

176 - "superconducting single-photon detectors." → superconducting single-photon detectors (SNSPDs).

213 - Is it possible to clarify how the author retrieved the fidelity ?

Conclusion :

285 - Summarization - I did not understand. Did the author want to say summation ?

SUPPLEMENTARY NOTE 1 :

_ number of photons p_j , we usually take n_j to avoid confusion with the probability notation.

_ Eq S17: what is ζ ?

SUPPLEMENTARY NOTE 2 :

_ Refer to S2 figure but the figure is S1

SUPPLEMENTARY NOTE 3 :

_ "and a central wavelength of 1550.116nm and a bandwidth of 1.9nm" → ", a central wavelength of 1550.116nm and a bandwidth of 1.9nm"

_ squeezing level of photons → Photons are not squeezed

_squeezed photons

_ What are the spiral lines ? Resonators ? Why is it so important that it is a spiral line ?

B. integrated photonic chip :

_ [0, 3]mI ?

_ squeezing photons

C. Detectors and Probabilistic number resolving :

_ detectors' → detector's

_ I did not really see how the author incorporate the efficiency in the probability formula

SUPPLEMENTARY NOTE 4: Device performance :

A. Source characterization

_ By using the word "FSR" did the author mean Free Spectral Range ? FSR is used for resonators not for MZI.

_ The filter is on the pump so it is not the generated squeezed state which is filtered or I may not understand.

_ degenerate SFWM it would be good if the author precise it in the main text

_ (fig S3) Squeeze parameter → Squeezing parameter

_ Squeezing in dB ?

_ "When one of the sources is connected directly to the detectors, the calculated squeezing value by second-order correlation measurements can reach around 0.6 when input power is around 0.55mW (Figure S3a)" : Could the author explain how the squeezing parameter is linked to the second-order correlation ?

C. Quantum characterization

_ "and they are always higher than 0.9 for two-photon clicks and higher than 0.8 for four-photon clicks." - If the fidelities are given in percent in the previous sentence, this fidelities

could also be in percent.

_ It would be good to compare fidelities obtained using the quantum circuit with fidelities obtained using a classical algorithm.

SUPPLEMENTARY NOTE 6: Data encoding details

_ What are r_1, r_2 ? Squeezing parameters ?

SUPPLEMENTARY NOTE 7: Error and scalability analysis

_ Why did the author place a polarization controller before the detector ?

SUPPLEMENTARY NOTE 8: Derivation of Non-Condon effects

_ no-Condon \rightarrow non-Condon

_ (3), they propose \rightarrow Cite the authors

I hope that these remarks can help the authors improve the article and assist the editor in providing clearer guidance on their decision.

Sincerely,

Manuscript ID: Nature Communications manuscript NCOMMS-23-34789-T

Paper title: Large-scale photonic network with squeezed vacuum states for molecular vibronic spectroscopy

Authors: H. H. Zhu, H. S. Chen, T. Chen, Y. Li, S. B. Luo, M. F. Karim, X. S. Luo, F. Gao, Q. Li, H. Cai, L. K. Chin, L. C. Kwek, B. Nordén, X. D. Zhang, and A. Q. Liu

Reply to Reviewer 1

We are grateful to the Reviewer for recognizing our innovation and contribution to the experimental achievement of large molecular simulation. We are happy to address the Reviewer's comments and thankful to the Reviewer for inspiring these valuable improvements.

Comment 1: *Nevertheless, I have a big concern about the quality of the manuscript. I often find mistakes and typos in the manuscripts and supplementary materials, and following the equation derivations in the supplementary material is difficult (and I suspect it would contain mistakes). I stopped reviewing the manuscript and could not proceed further. Authors should improve their manuscript significantly for correctness and clarity. Please recheck every detail.*

Answer 1: As suggested by the Reviewer, we diligently addressed all errors and typographical issues in the manuscript, especially the supplementary materials, and carefully checked all mistakes, ensuring the manuscript now flows smoothly and is readily understandable.

Overall, we have made significant revisions to the manuscript. The main changes include:

1. SUPPLEMENTARY NOTE 1: Theory on vibronic spectra with squeezed vacuum state (Page 2 of Supplementary material) is revised to include a comprehensive definition of all symbols used in formulas with detailed derivations.
2. Mistakes and incorrect descriptions regarding molecules and mechanisms in vibronic spectra are rectified. More precise descriptions of experimental results (Page 8) are provided with details of the molecules and an analysis of experimental errors.
3. A new experiment on vibrational excitation simulation is added (Page 11 of Supplementary material) to demonstrate further the broad range of application scenarios for our model.
4. A detailed comparison with previous works and the original approach (Page 10) is included to highlight the advantages of our simulation method for molecular vibronic spectra.

We hope that the revised manuscript is acceptable to the Reviewer and meets the quality of Nature Communications.

Reply to Reviewer 2

We are grateful to the Reviewer for recognizing our contribution to the experimental achievement of simulating Franck-Condon factors of vibronic spectra. We are happy to address the Reviewer's comments and thankful to the Reviewer for inspiring these valuable improvements.

Comment 1.1: *Firstly, could the authors state clearly how they obtained the frequencies and Duschinsky rotation matrices used in this paper? These values are crucial inputs for the Franck-Condon model, and the accuracy of spectra is only as good as the underlying accuracy of the frequencies/rotations that define the potential.*

Answer 1.1: As suggested by the Reviewer, in the Supplementary Material for Ref. [35], the equilibrium structure, **frequencies**, and mass-weighted normal coordinates of the initial and final state for both **formic acid** and **thymine** are given. One can obtain the **Duschinsky rotation matrices** by integrating the corresponding mass-weighted normal coordinates of the initial and final state.

A detailed description of Duschinsky rotation is added in **SUPPLEMENTARY NOTE 1** on Page 2 of Supplementary material:

“On the other hand, within the Born-Oppenheimer approximation and the Condon approximation, the intensity of vibronic transition is the square of the overlap integral between the relative vibrational wave functions of different potential energy surfaces (Fig. 1b), which is known as Frank-Condon factors (FCF). The coordinate space representations of the vibrational states are obtained by projecting on the position operator eigenstates $|Q\rangle$ and $|Q'\rangle$, respectively. $|Q\rangle$ and $|Q'\rangle$ are the dimensionless normal coordinates of the initial and final states, which are defined by

$$R - R_0 = \hbar^{1/2} M^{1/2} L \Omega^{-1/2} Q \quad [\text{S13a}]$$

$$R' - R'_0 = \hbar^{1/2} M'^{1/2} L' \Omega'^{-1/2} Q' \quad [\text{S13b}]$$

where R and R' denote the Cartesian displacement coordinates with respect to the reference configurations coinciding with the equilibrium structures R_0 and R'_0 , respectively, both of which have the same Eckart axis system attached. L and L' denote the normal modes of the initial and final states analyzed in the mass-weighted Cartesian coordinates. Ω and Ω' are the diagonal matrix of harmonic oscillator frequencies and M is the diagonal matrix of atomic masses.

When transformation between the Eckart axis systems of the final and initial states is an identity, the relation between the initial and final dimensionless normal coordinate is expressed as

$$\begin{aligned} \hat{Q}' &= \Omega'^{1/2} L'^t L \Omega^{-1/2} \hat{Q} + \hbar^{-1/2} \Omega'^{1/2} L'^t M^{1/2} \Delta R \\ &= J \hat{Q} + \delta \\ &= \hat{U}_{Dok}^\dagger \hat{Q} \hat{U}_{Dok} \end{aligned} \quad [\text{S15}]$$

where $\Delta R = R_0 - R'_0$ is the difference of equilibrium geometries between the initial and final states in the Cartesian coordinates, $L'^t L$ is the Duschinsky matrix and δ is the displacement

vector responsible for the molecular structural changes along the normal coordinates. \hat{U}_{Dok} is the Doktorov transformation between the relative potential energy surfaces (PESs) of the molecule, which can be decomposed as $\hat{U}_{Dok} = \hat{D}_\alpha \hat{R}_U \hat{S}_r \hat{R}_{U'}$, where $U \text{diag}(\mathbf{r})U' = J$ is the singular value decomposition (SVD) of J and $\alpha = \frac{1}{\sqrt{2}} \delta$.”

Comment 1.2: *Notably, the authors incorrectly state that formic acid has 7 vibrational modes. The reduction to 7 modes is confusing considering the fact that the peaks in the spectrum are captured in a 4-mode model. The authors then correctly state that thymine has 39 vibrational modes, but only provide parameters for a 7-mode model (which only truly includes 6 modes if the leading value of δ is zero). There is no explanation in the manuscript how the authors chose which modes to use in the reduced models, and in the case of thymine there is no indication in the main text that the reduction was made.*

Answer 1.2: As pointed out by the Reviewer, the formic acid should be 9 modes as shown in Ref [10]. Our earlier statement erroneously claimed that formic acid has only 7 vibrational modes, which corresponds to the symmetry block part a' of the molecule. In the revised manuscript, we have corrected the incorrect description of formic acid's vibrational modes and chosen 4 modes as the optimal selection for our experiment. For thymine, it has 39 vibrational modes, and we specifically chose 7 modes as the optimal selection in our experiment. These selection rules are primarily guided by the Franck-Condon factors, with preference given to those with larger values due to the constraints imposed by the total chip size. For thymine, the mode with a displacement of 0 was selected because the 2-photon event of this mode is large, although its 1-photon event is 0. This information is discussed in the revised manuscript.

“The quantum photonic chip is employed to compute the FC profiles of two molecules: formic acid (CH_2O_2) and thymine ($\text{C}_5\text{H}_6\text{N}_2\text{O}_2$), which possess a considerable number of vibrational modes, specifically 9 modes for formic acid and 39 modes for thymine. Due to the size limitations of the chip, we deliberately exclude certain modes with relatively small Franck-Condon factors. For comprehensive information on the chosen vibrational frequencies and local basis transformations in our experiment, please refer to **SUPPLEMENTARY NOTE 5**.” in Line 30 on Page 7.

“For formic acid (CH_2O_2) with 9 vibrational modes, in the experiments, only 4 quantum harmonic oscillators are selected and then effectively doubled and encoded to an 8-mode network. The incoming photons from 1st- and 5th-modes are probabilistically split into four waveguides via three 1×2 beam-splitters.” in Line 9 on Page 8.

“Calculating the FC profile of thymine ($\text{C}_5\text{H}_6\text{N}_2\text{O}_2$) presents a more intricate task due to its vibronic boson sampling involving 39 modes. To manage this complexity, we chose to work with a subset of 7 modes, which have been meticulously encoded into our chip. Furthermore, the selected 1st-, 6th-, and 7th-modes are probabilistically split into two waveguides via a 1×2 beam-splitter.” in Line 25 on Page 8.

“Here, we include all computed characteristic parameters, such as vibrational frequencies, displacement vector and transformation matrices used in the work.

A. Formic Acid

$$U = \begin{pmatrix} 0.8423 & 0.1799 & -0.3857 & 0.3074 \\ -0.3403 & -0.5231 & -0.6679 & 0.3848 \\ -0.4004 & 0.7636 & -0.1036 & 0.4838 \\ -0.0907 & 0.3151 & -0.5900 & -0.7193 \end{pmatrix}$$

$$\omega = \begin{pmatrix} 1825.1799 \\ 1416.9512 \\ 1326.4684 \\ 1137.04 \end{pmatrix}, \omega' = \begin{pmatrix} 1566.4602 \\ 1399.6554 \\ 1215.3421 \\ 1190.9077 \end{pmatrix}, \delta = \begin{pmatrix} 1.5599 \\ -0.3784 \\ 0.4553 \\ -0.3439 \end{pmatrix}$$

B. Thymine

$$U = \begin{pmatrix} -0.0373 & 0 & 0 & 0 & 0 & 0 & 0 \\ 0 & -0.3441 & 0.0318 & -0.2303 & -0.1322 & 0.162 & 0.1761 \\ 0 & 0.2715 & -0.1283 & 0.2791 & 0.4629 & 0.0143 & 0.2017 \\ 0 & 0.2184 & -0.262 & 0.2368 & -0.2354 & 0.0205 & -0.1328 \\ 0 & 0.0153 & -0.2638 & -0.1924 & -0.1226 & 0.0199 & -0.1291 \\ 0 & -0.1801 & 0.1629 & 0.1492 & -0.063 & -0.0567 & 0.4707 \\ 0 & -0.1157 & -0.1349 & 0.1014 & -0.1531 & 0.1296 & 0.0048 \end{pmatrix}$$

$$\omega = \begin{pmatrix} 108.354 \\ 390.234 \\ 547.949 \\ 732.620 \\ 1219.420 \\ 1372.674 \\ 1697.986 \end{pmatrix}, \omega' = \begin{pmatrix} 57.567 \\ 392.319 \\ 532.098 \\ 690.284 \\ 1270.82 \\ 1314.60 \\ 1574.93 \end{pmatrix}, \delta = \begin{pmatrix} 0 \\ -0.7399 \\ -0.6850 \\ 0.6530 \\ 0.4817 \\ 0.8490 \\ -0.8732 \end{pmatrix}$$

in Line 21 on Page 14 of the Supplementary Materials.

Comment 2.1: Based on the raw data, it appears that the authors binned frequencies in the spectra in intervals of 50 cm^{-1} . It is not clear why binning is done at all. It leads to extra data handling, since the raw experimental data is based on output frequency and photon count. It also leads to a clear misalignment of peak positions, and mischaracterization of overlapping peaks as a single peak.

Answer 2.1: As pointed out by the reviewer, the binned frequencies are meaningless and lead to extra misalignment. Therefore, we have removed the unnecessary data handling and directly retained the molecule's vibrational frequencies as the x -axis coordinates. Figure 3 is revised as:

Fig. 3 Vibronic spectra reconstruction. Franck–Condon profiles are obtained from chip distributions programmed according to the vibronic transitions of formic acid (a, with the structure shown in the inset) and thymine (b, with the structure shown in the inset). The grey bar graphs depict the histogram of energies, whereas red and blue continuous curves show the Lorentzian broadening of the bars.

Comment 2.2: For example, comparing Fig. 3a in this manuscript to Fig. 3 in ref. 10 shows a clear difference despite identical input parameters. The thymine spectrum shows low-frequency peaks that are not possible if the inputs in the supporting information are correct.

Answer 2.2: As pointed out by the Reviewer, in Fig. 3a, the primary distinction is observed at high frequencies. In line with the current experimental results obtained using a 16-mode quantum chip and considering the maximum of four-photon events achieved, the experimental FC value is lower than the theoretical values in the high-frequency region.

In Fig. 3b, the theoretical expectation is a zero peak at low frequency ($\omega' = 58$), representing the mode with zero displacement. Nevertheless, our experimental results reveal the presence of a peak. This disparity arises from the excessively high single-photon events associated with this mode recorded experimentally, stemming from errors originating from several factors. First, for this frequency ($\omega' = 58$), additional multiphoton events have been erroneously recorded as single photons despite implementing a two-photon number resolving approach. Second, due to photon loss, certain multiphoton events are mistakenly categorized as single photons, thereby inflating the count of single-photon events. Third, the influence of various noise effects should be considered.

Such differences are discussed in the revised manuscript.

“In particular, the loss obviously affects the FC factor values at high frequencies. Because of the photon loss, only a maximum of four-photon coincidence is achievable. Therefore, events with higher than four photons cannot be stored and further reduce the FC factor values at high frequency (see a more detailed discussion in **SUPPLEMENTARY NOTE 7**.” in Line 20 on Page 8.

“In addition, photon loss, leading to the misclassification of certain multiphoton events as single photons, inflates the count of single-photon events. A detailed discussion about these errors is in **SUPPLEMENTARY NOTE 7**.” in Line 5 on Page 9.

A detailed error analysis is shown in “**SUPPLEMENTARY NOTE 7**” as:

“Besides enhancing the chip performance, photon loss calibration and increased photon number resolution can also reduce errors. Within the theoretical framework, one-photon events are defined as a_i (where i denotes the mode label), two-photon events as $b_{i,j}$ (with i and j as mode labels), three-photon events as $c_{i,j,k}$ (with i , j , and k as mode labels), and four-photon events as $d_{i,j,k,l}$ (with i , j , k , and l as mode labels). To account for photon loss, we introduce a loss parameter η . Consequently, the experimentally obtained one-photon event, denoted as a'_i , can be expressed as

$$a'_i = (1 - \eta)a_i + \eta(1 - \eta) \sum_j b_{i,j} + \eta^2(1 - \eta) \sum_{j,k} c_{i,j,k} + \eta^3(1 - \eta) \sum_{j,k,l} d_{i,j,k,l}$$

Similarly, we can establish the relationship between experimental and theoretical two-photon, three-photon, and four-photon events, and so forth. This equation demonstrates that experimentally obtained one-photon events encompass not only the primary one-photon events but also the components lost from multi-photon events. After correction, this source of error can be mitigated. Furthermore, increasing the resolution of the photon number counting can reduce the likelihood of erroneously classifying a multiphoton event as a single-photon event.” on Page 19

Comment 3.1: *The authors state that "Ref. [27] showed that inharmonicities could be considered and implemented on linear optical quantum hardware". This appears to be a misunderstanding, since ref. 27 and the work done in this manuscript are for harmonic models. Rather, the corresponding work includes the non-Condon, Herzberg-Teller couplings.*

Answer 3.1: As pointed out by the Reviewer, the anharmonic and non-Condon are two different effects and mechanisms in vibronic spectra. In particular, the non-Condon effect (or Herzberg-Teller approximation) is due to the consideration of coordinate dependence of the transition dipole moment and the neglect of higher-order terms. The incorrect statement is revised as

“The Condon approximation, which assumes a constant transition dipole moment (TDM), may no longer suffice in simulating some molecules. To improve the accuracy, it is necessary to include the non-Condon approach. Ref. [33] shows that a non-Condon approximation could be considered and implemented on linear optical quantum hardware.” in Line 12 on Page 9.

Comment 3.2: *How did the authors obtain the Herzberg-Teller coupling parameters used in this work?*

Answer 3.2: As pointed out by the Reviewer, to incorporate the non-Condon effect, we start from the Taylor expansion of the transition dipole moment (TDM) operator. The coefficients found in the expansion of the TDM operator are denoted as μ . ω and ω' are the harmonic frequencies of the initial and excited states, respectively. U_D is the Duschinsky unitary matrix, and d is the displacement vector. An exhaustive derivation of the non-Condon effects are presented in **Supplementary Note 8**.

“However, some vibration process of chemical molecules involves Herzberg-Teller (HT) couplings, which render the Franck–Condon (FC) approximation inadequate, and the non-Condon effect should be considered. In such a case, the TDM is not a constant. By employing the Taylor expansion of the TDM operator ($\hat{\mu}$) with respect to the normal coordinates of molecules, known as the HT expansion, we can derive the first-order HT expansion when $\hat{\mu}$ is expanded up to the linear term.

$$\hat{\mu} = \mu^{(0)} + \sum_{i=1}^m \mu_i^{(1)} \hat{Q}_i + \dots \quad [\text{S38}]$$

where $\mu_i^{(1)}$ is the first derivative with respect to the i th normal coordinate (Q_i) at the equilibrium structure of the initial state.” on Page 21

Comment 3.3: *Later, the authors state that "Although the classical computation of the FC profile is easier by exploiting molecular symmetry, molecules often are asymmetry, especially in the case of large molecules, in which non-Condon effects should be considered". What is the relation between non-Condon effects and asymmetry?*

Answer 3.3: As pointed out by the Reviewer, there is no discernible connection between non-Condon effects and asymmetry. The statement is removed and replaced in the revised manuscript as

“While the classical computation of the FC profile is easier by relying on the Condon approximation, it is essential to acknowledge that, for many large systems, FC-forbidden or weakly FC-allowed transitions cannot be excluded, and the FC assumption is known to break down in some cases (most notably in aromatic molecules). In such instances, it becomes imperative to introduce corrections to ensure a meaningful interpretation of the observed transitions.” in Line 11 on Page 9.

Comment 4.1: *Finally, the authors lack any discussion of their work in comparison to past results. The stated purpose is to make boson sampling practical for larger systems, but how does it compare to the work cited in this manuscript?*

Answer 4.1: As suggested by the Reviewer, the comparison with previous works is added in the revised manuscript.

“Utilizing the theoretical model in conjunction with the experimental hardware platform, our chip exhibits a remarkable capability to simulate the FCP of larger molecules with 7 modes, surpassing prior research efforts. Our research reveals noteworthy distinctions from prior studies, where previous works were limited to simulating vibronic spectra exclusively for virtual molecules [20, 31] or molecules featuring only two modes, such as tropolone [34], SO₂ [13, 14], H₂O [12], and O₃ [12]. It is worth noting that experiments involving molecules with displacement are yet to be demonstrated in optical platforms. In contrast, for the first time, we demonstrated the vibronic spectra of real molecules comprising 7 modes for thymine while keeping the displacement, underscoring the significant potential of our model in the realm of molecular simulations for larger and more complex systems.” *in Line 1 on Page 11.*

Comment 4.2: *Notably, the Franck-Condon model fails for many large systems where vibronic coupling and environmental noise are prevalent (cf. <https://doi.org/10.1039/D3SC02453A>). Although the authors have written the manuscript in terms of photoelectron spectra (which is not necessarily the only use of Franck-Condon factors), it should be noted that for many molecules this approach would at best be applicable for the photoelectron spectrum of the ground electronic state of the cation. For some molecules (cf. <https://doi.org/10.1039/B500055F>) even the ground cationic state has strong vibronic coupling.*

Answer 4.2: As suggested by the Reviewer, FC approximation offers a simple, intuitive model for understanding electronic spectra. FC approximation is both adequate and sufficiently accurate for assigning spectral lines in many molecular systems [32]. However, the FC assumption is not always sufficiently accurate for assigning spectral lines in molecular systems and vibronic coupling has to be considered. In this manuscript, we added the non-Condon effects and experimentally demonstrated the possibility of simulating the non-Condon effects in molecular spectroscopy using the quantum chip, as shown in the “**Non-Condon effects in molecular spectroscopy**” section. In addition, as mentioned by the Reviewer, photoelectron spectra are not the only use of FC factors. We further add some discussion about the vibrational excitation simulation in the revised manuscript.

“In addition, within the realm of molecular simulation, FC factors serve as a valuable tool not only for exploring molecular vibronic spectra but also for simulating molecular vibrational excitations during vibronic transitions [35]. A detailed discussion is added in **SUPPLEMENTARY NOTE 9.**” *in Line 12 on Page 11.*

“**SUPPLEMENTARY NOTE 9: Vibrational excitation simulation**”

During molecular simulation, FC factors are used not only in studying the molecular vibronic spectra but also for other simulations, such as molecular vibrational excitations during vibronic transitions [7] and single-molecule electron transport [8]. Figure S6 depicts the experimental demonstration of vibrational excitations in pyrrole during photoexcitation-mediated vibronic processes. For instance, we examine the vibrational excitation of six out of 24 normal modes in pyrrole's excited electronic state, both with and without pre-excitation of the oxygen-hydrogen (O-H) stretching mode at the ground electronic state (detailed in **SUPPLEMENTARY NOTE 5**). From the marginal distributions, the initial pre-excitation is transferred to two vibrational modes of the excited electronic state, with the vibrational frequencies of 3231.5 cm⁻¹ and 3250.8 cm⁻¹ due to the Duschinsky mode mixing. This proof-of-principle demonstration shows the

possibility of predicting the vibrational excitation of molecules in vibronic processes using our quantum microprocessor, which is also used for various molecular simulations in the future.”

Fig. S6: Vibrational excitation simulations. Single-mode marginal distribution of pyrrole during a vibronic transition from ground to the first excited state without (top) and with (bottom) pre-excitation of the ground state O–H stretching mode of pyrrole.

Comment 5: *On line 79, "the investigation of bigger molecule sizes is constrained by the weak coupling of next-nearest neighbors among ions and superconducting qubits". Whereas superconducting qubits are typically only coupled to nearest neighbors, trapped ions have global coupling through trap vibrational modes.*

Answer 5: As pointed out by the reviewer, indeed, mode coupling is not the constraining factor for bigger molecule sizes. The core problem is that most superconducting and ion-trap computers are too susceptible to noise and low computational fidelity [1, 2]. Most superconducting quantum computers are too error-limited to produce dependable quantitative results for sufficiently large (six spins or more) systems. Postprocessing techniques are often required to mitigate errors in the raw results to achieve quantitative chemical accuracy for bonding energies [3, 4]. Ions in two-dimensional traps are sufficiently isolated from the environment, leading to excellent coherence times and interaction with radio frequency. They possess high-fidelity gate operations and long coherence times. However, many ions are still plagued by gate error sources like anomalous heating from the surface effects of electrodes and crosstalk. To date, only a few molecules with few atoms have been successfully demonstrated on this architecture [5, 6]. Therefore, integrated photonic chips with Gaussian boson sampling offer an alternative viable alternative for the realization of larger molecules for quantum chemistry.

- [1] Lau J W Z, Lim K H, Shrotriya H, et al. NISQ computing: where are we and where do we go? AAPPs Bulletin, 2022, 32(1): 27.
- [2] Bharti K, Cervera-Lierta A, Kyaw T H, et al. Noisy intermediate-scale quantum algorithms. Reviews of Modern Physics, 2022, 94(1): 015004.
- [3] O'Malley P J J, Babbush R, Kivlichan I D, et al. Scalable quantum simulation of molecular energies. Physical Review X, 2016, 6(3): 031007.

- [4] Kandala A, Mezzacapo A, Temme K, et al. Hardware-efficient variational quantum eigensolver for small molecules and quantum magnets. *Nature*, 2017, 549(7671): 242-246.
- [5] Hempel C, Maier C, Romero J, et al. Quantum chemistry calculations on a trapped-ion quantum simulator. *Physical Review X*, 2018, 8(3): 031022.
- [6] Nam Y, Chen J S, Pienti N C, et al. Ground-state energy estimation of the water molecule on a trapped-ion quantum computer. *npj Quantum Information*, 2020, 6(1): 33.

The statement is revised in the revised manuscript as

“Yet, the study of larger molecules is hampered by the susceptibility of superconducting and trapped ion systems to various sources of noise and limited gate fidelity [15, 16].” in Line 26 on Page 3.

- 33. Lau J W Z, Lim K H, Shrotriya H, et al. NISQ computing: where are we and where do we go? *AAPPS Bulletin*, 2022, 32(1): 27.
- 34. Bharti K, Cervera-Lierta A, Kyaw T H, et al. Noisy intermediate-scale quantum algorithms. *Reviews of Modern Physics*, 2022, 94(1): 015004.

Comment 6: *On line 117, "the square of the overlap integral between the relative vibrational wave functions of different potential energy surfaces within the Born-Oppenheimer approximation". This formulation is within the Born-Oppenheimer approximation and the Condon approximation, which is an important distinction considering the part of the manuscript about non-Condon effects.*

Answer 6: As pointed out by the reviewer, the Condon approximation is added in the revised manuscript as

“Photoelectron spectroscopy describes the molecular electronic transition from a neutral state to a cationic state, and the spectral profile can be obtained by computing the corresponding Franck-Condon factors (FCF), which is the square of the overlap integral between the relative vibrational wave functions of different potential energy surfaces within the **Born-Oppenheimer approximation and the Condon approximation** [24 -26].” in Line 1 on Page 5.

- 26. Hazra A, Nooijen M. Comparison of various Franck–Condon and vibronic coupling approaches for simulating electronic spectra: The case of the lowest photoelectron band of ethylene. *Physical Chemistry Chemical Physics*, 2005, 7(8): 1759-1771.

Comment 7: *Fig. 2 has several abbreviations that aren't explained in the main text or the caption.*

Answer 7: As pointed out by the reviewer, the caption of Fig. 2 is revised in the revised manuscript.

Fig. 2 Schematic of a quantum microprocessor chip and experimental setup. **a:** Setup to generate single-mode squeezing using dual pumps from 2-ps laser pulses, consisting of an optical pulse compressor, an erbium-doped fiber amplifier (EDFA), wavelength-division multiplexers (WDM), and a polarization controller (PC). The two pumps at the wavelength of 1546 nm and 1553 nm were selected and combined using WDM and coupled into the chip through a grating coupler. **b:** Photograph of microprocessor chip. The spiral sources produce a vacuum-squeezed state; the programmable interferometer network achieves an arbitrary unitary matrix; and on-chip beam splitters are used for pseudo-number-resolving detection. The single pump input light is coupled to the chip by a one-dimensional subwavelength grating coupler. Output photons are coupled by edge couplers to fiber arrays. The scale bar denotes 100 μm . **c:** An overview of off-chip control devices and measurement, including temperature controller (TEC), electrical modulators, output photon detection (Time Tagger), Analog-to-Digital Converter (DAQ), and data processing with a server computer.

Comment 8: Please clarify the sentence on line 145, "the probability amplitudes of squeezed vacuum state and linear network are quadratic to Hafnian function as ..."

Answer 8: As pointed out by the reviewer, the statement is modified in the revised manuscript as

“Meanwhile, the probability amplitudes of a photon pattern are proportional to the square of the Hafnian function as $\Pr(\mathbf{p}) = \frac{1}{p! \sqrt{\sigma_Q}} |\text{haf}(A^{\mathbf{p}})|^2$.” in Line 4 on Page 6.

Comment 9: *On line 150, "fork basis" should be Fock basis*

Answer 9: As pointed out by the reviewer, the typo is corrected in Line 6 on Page 6 of the revised manuscript.

Comment 10: *On line 228, "two complex molecules with a point-group symmetry". I would hesitate to call either molecule complex, and all molecules have a point group symmetry of at least C1.*

Answer 10: As pointed out by the reviewer, the word “complex” is deleted in the revised manuscript as

“In conclusion, the vibronic spectroscopies of two molecules with a point-group symmetry are successfully simulated experimentally on the quantum photonic chip.” in Line 10 on Page 9.

Comment 11: *Why do parameters for pyrrole appear in the supporting information?*

Answer 11: Our revised manuscript includes a detailed description of vibrational excitation simulation in **SUPPLEMENTARY NOTE 9**, focusing on using pyrrole as the target molecule. The parameters of pyrrole are added to **SUPPLEMENTARY NOTE 5**.

In summary, all reviewer comments are addressed. In addition, the manuscript and supplementary materials are carefully proofread.

Reply to Reviewer 3

We are grateful to the Reviewer for recognizing our contribution to demonstrating the advance in quantum simulation. We are happy to address the Reviewer's comments and thankful to the Reviewer for inspiring these valuable enhancements.

Comment 1: *The authors state the displacement operation as an experimental challenge to emphasize the simplicity of the presented algorithm; however, it seems to me that incrementing the number of single-mode squeezed vacuum with the exactly the same squeezing parameter is more challenging than implementing a linear operation of displacement. The authors could explain more specifically why this algorithm is easier to implement.*

Answer 1: As suggested by the Reviewer, a detailed discussion of the implementation is provided. The generation of single-mode squeezed vacuum states is made possible using spirally shaped waveguides, capitalizing on spontaneous four-wave mixing (SFWM) effects. Within our integrated quantum chip, the expansion of the squeezed vacuum state can be achieved by incrementally adding spirally shaped waveguides. Notably, the unique advantages offered by silicon photonic integrated circuits, including ultra-compact size and high-density integration, render the scaling of squeezed vacuum state sources a straightforward technological endeavor. Furthermore, the modulation of the squeezed parameter can be effortlessly accomplished by introducing a Mach-Zehnder interferometer (MZI) to adjust the pump intensity in each mode.

Nonetheless, achieving displacement involves the interference between non-classical squeezed light and laser light. Challenges arise during this process, primarily in managing the split ratio between the squeezed and coherent states. When the laser intensity significantly surpasses that of the squeezed light, it becomes possible to approximate the output photon distribution classically. While this may enable efficient sampling using classical algorithms, it precludes the realization of a quantum computational advantage.

Designing a splitting device with both precision and a substantial split ratio proves to be a formidable task within an integrated platform, and we have yet to witness the successful creation of a coherent squeezed state on a chip. When contemplating the need for source design in bulk optics, it necessitates the incorporation of weighty plates, lenses, and modulation equipment (for phase locking), all of which consume significant space. This limitation poses a substantial hurdle to achieving large-scale implementation.

In summary, the generation and scalability of single-mode squeezed vacuum states is easier than that of squeezed coherent states. The integration benefits of silicon photonic circuits make scaling this squeezed vacuum state straightforward. However, the challenges of managing the split ratio between coherent and squeezed states on an integrated chip hinder the large-scale implementation of squeezed coherent states.

We revised the related part in the manuscript as

“A commonly employed approach involves generating interference between squeezing and coherent states, necessitating a precise and substantial split ratio, typically accomplished through bulk optics [22, 23]. However, the practical implementation of a modulated coherent state and squeezed vacuum state in bulk optics presents significant challenges, entailing bulky components

such as plates, lenses, and modulation equipment for phase locking, all occupying considerable space [22]. Consequently, the experimental realization of large-scale molecular vibronic simulations remains a formidable undertaking by employing the squeezed coherent states algorithm.” in Line 10 on Page 4.

Comment 2: *The reconstructed fidelity for vibronic spectra of naphthalene, phenanthrene and benzene under non-Condon regime is not reported in the paper. Is there a specific reason I missed it ?*

Answer 2: As suggested by the Reviewer, the reconstructed fidelities for vibronic spectra of naphthalene, phenanthrene, and benzene under the non-Condon regime are added in the revised manuscript as

“The measured fidelities with the vibronic spectra of naphthalene, phenanthrene, and benzene are 86.3%, 95.3% and 99.4%, respectively.” in Line 30 on Page 9.

Comment 3: *Moreover, it should be good if the author show a significant benchmark against classical simulation using this criteria : $C = F_Q - F_C$ the difference between the fidelity of the reconstructed Franck-Condon profile F_Q and the optimal fidelity obtainable from a classical strategy F_C issued from the following paper the author should cite :*

<https://www.nature.com/articles/s41567-019-0567-8>

<https://iopscience.iop.org/article/10.1088/1361-6455/aaf031>

Answer 3: As suggested by the Reviewer, this criterion is added to compare the difference between quantum simulation and classical simulation in the revised manuscript as

“We further calculate the difference $C = F_Q - F_C$ between the fidelity of the reconstructed FC profile F_Q and the optimal fidelity obtainable from a classical strategy F_C to benchmark against a classical simulation [31, 32]. A small improvement over a classical strategy ($C = 1.7\%$) is obtained for formic acid molecules.” in Line 21 on Page 8.

“The fidelity reaches 98.3%, corresponding to $C = 1.3\%$.” is added in Line 1 on Page 9.

[31] Paesani S, Ding Y, Santagati R, et al. Generation and sampling of quantum states of light in a silicon chip. *Nature Physics*, 2019, 15(9): 925-929.

[32] Clements W R, Renema J J, Eckstein A, et al. Approximating vibronic spectroscopy with imperfect quantum optics. *Journal of Physics B: Atomic, Molecular and Optical Physics*, 2018, 51(24): 245503.

Comment 4: I however have one concern about the source characterization in the supplementary note 4 A. Source characterization. The authors claimed they are able to determine the squeezing value of their source by second-order order correlation measurement. This part needs to be clarified, and authors should explain how they retrieve the squeezing value of the source with this measurement.

Answer 4: As suggested by the Reviewer, the measurement of squeezing value from two-photon correlations using the threshold detectors in the experiment is added in the Supplementary Materials as

“The source emission's single-mode squeezing (SMS) parameter is calculated using two-mode second-order correlation measurements in the experiment setup schematized in Figure 2. Ref. [3] reported the calculation of the two-mode squeezing (TMS) parameter of the source using two-mode second-order correlation measurements. Our method is similar to this by changing the source state to SMS state instead of TMS. Given a pure single-mode SMS state, the probability of detecting n_s (n_i) single photons in the signal (idler) mode in the loss-less case is expressed as

$$p_{SMS}(n_s, n_i) = \frac{\tanh^{2n_s}(\xi) (2n_s)!}{\cosh(\xi) 2^{2n_s} n_s! n_s!} \delta_{n_s, n_i} = p_0(n_s) \delta_{n_s, n_i} \quad (S25)$$

In the presence of losses, the detection probability is given by

$$\begin{aligned} p(n_s, n_i) &= \left(\frac{\eta}{1-\eta}\right)^{n_s+n_i} \sum_{n \geq \max\{n_s, n_i\}} p_0(n) (1-\eta)^{2n} \\ &= A(\xi, \eta) \left(\frac{\eta}{1-\eta}\right)^{n_s+n_i} \sum_{n \geq \max\{n_s, n_i\}} \frac{\tanh^{2n}(\xi) (2n)!}{2^{2n} n! n!} (1-\eta)^{2n} \\ &= A(\xi, \eta) \left(\frac{\eta}{1-\eta}\right)^{n_s+n_i} \sum_{n \geq \max\{n_s, n_i\}} \tanh^{2n}(\xi) \frac{1}{\sqrt{\pi n}} (1-\eta)^{2n} \\ &= A(\xi, \eta) \left(\frac{\eta}{1-\eta}\right)^{n_s+n_i} \sum_{n \geq \max\{n_s, n_i\}} (a_1 b_1^{2n} + a_2 b_2^{2n} + a_3 b_3^{2n}) \tanh^{2n}(\xi) (1-\eta)^{2n} \\ &= A(\xi, \eta) \left(\frac{\eta}{1-\eta}\right)^{n_s+n_i} \left\{ \begin{aligned} &a_1 \left(\sum_{n=0}^{\infty} [b_1 \tanh(\xi) (1-\eta)]^{2n} - \sum_{n=0}^{\max\{n_s, n_i\}} [b_1 \tanh(\xi) (1-\eta)]^{2n} \right) \\ &+ a_2 \left(\sum_{n=0}^{\infty} [b_2 \tanh(\xi) (1-\eta)]^{2n} - \sum_{n=0}^{\max\{n_s, n_i\}} [b_2 \tanh(\xi) (1-\eta)]^{2n} \right) \\ &+ a_3 \left(\sum_{n=0}^{\infty} [b_3 \tanh(\xi) (1-\eta)]^{2n} - \sum_{n=0}^{\max\{n_s, n_i\}} [b_3 \tanh(\xi) (1-\eta)]^{2n} \right) \end{aligned} \right\} \\ &= B(\xi, \eta) \left(\frac{\eta}{1-\eta}\right)^{n_s+n_i} \left\{ \frac{a_1}{q_1(\xi, \eta)} [b_1 \tanh(\xi) (1-\eta)]^{2\max\{n_s, n_i\}} \right. \\ &\quad \left. + \frac{a_2}{q_2(\xi, \eta)} [b_2 \tanh(\xi) (1-\eta)]^{2\max\{n_s, n_i\}} + \frac{a_3}{q_3(\xi, \eta)} [b_3 \tanh(\xi) (1-\eta)]^{2\max\{n_s, n_i\}} \right\} \end{aligned} \quad (S26)$$

where $\{a_k, b_k\} = \{0.5913, 0.4883\}; \{0.2705, 0.9243\}; \{0.1357, 0.995\}$, $q_i = 1 - b_i \tanh(\xi) (1-\eta)$ ($i = 1, 2, 3$) and $A(\xi, \eta)$ and $B(\xi, \eta)$ are normalization constants.

To determine the constant $B(\xi, \eta)$, the normalization condition $\sum_{n_s, n_i=0}^{\lim} p(n_s, n_i) = 1$ is imposed and rewrite as

$$\sum_{n_s, n_i=0}^{\lim} p(n_s, n_i) = 1 = 2 \sum_{n_i=0}^{\lim} \sum_{n_s \geq n_i} p(n_s, n_i) - \sum_{n_i=0}^{\lim} p(n, n) \quad (S27)$$

and

$$\frac{a_1 B(\xi, \eta)}{\xi(a)(1-\eta^2 b_1^2 \tanh^2 \xi) q_1(\xi, \eta)} + \frac{a_2 B(\xi, \eta)}{\xi(b)(1-\eta^2 b_2^2 \tanh^2 \xi) q_2(\xi, \eta)} + \frac{a_3 B(\xi, \eta)}{\xi(c)(1-\eta^2 b_3^2 \tanh^2 \xi) q_3(\xi, \eta)} = 1 \quad (\text{S28})$$

where $a = \eta(1-\eta)\tanh^2 \xi b_1^2$, $b = \eta(1-\eta)\tanh^2 \xi b_2^2$, $c = \eta(1-\eta)\tanh^2 \xi b_3^2$ and $\xi(x) = (1-x)/(1+x)$.

Then, correlation measurements at time delay $\Delta t = 0$ thus represent events whenever $n_s \geq 1$ signal photons and $n_i \geq 1$ idler photons from a single SMS arrive at the detectors simultaneously, which happens with a probability

$$\begin{aligned} p_{\text{coinc}}(\Delta t = 0) &= p(n_s \geq 1, n_i \geq 1) \\ &= 1 - p(0, 0) - p(0, 1) - p(1, 0) \\ &= 1 - \left(\frac{a_1}{q_1} + \frac{a_2}{q_2} + \frac{a_3}{q_3} \right) B - 2 \left(\frac{a_1 b_1^2}{q_1} + \frac{a_2 b_2^2}{q_2} + \frac{a_3 b_3^2}{q_3} \right) B \tanh^2(\xi) \eta(1-\eta) \end{aligned} \quad (\text{S29})$$

The probability of having this event with a time delay $\Delta t > 0$ is

$$\begin{aligned} p_{\text{coinc}}(\Delta t > 0) &= p(n_s \geq 1, n_i \geq 1) \\ &= \left(\sum_{n_i=0}^{\text{lim}} p(n_s \geq 1, n_i) \right) \left(\sum_{n_s=0}^{\text{lim}} p(n_s, n_i \geq 1) \right) \\ &= \left(1 - \frac{a_1 B}{q_1(1-a)} - \frac{a_2 B}{q_2(1-b)} - \frac{a_3 B}{q_3(1-c)} \right)^2 \end{aligned} \quad (\text{S30})$$

Then, the quantity to be considered is the ratio between the coincidences measured at $\Delta t > 0$ and those measured at $\Delta t = 0$,

$$R = \frac{p_{\text{coinc}}(\Delta t > 0)}{p_{\text{coinc}}(\Delta t = 0)} \quad (\text{S31})$$

In our experiment, the squeezing parameters are set small, and losses in the detection channels are significant. Under this approximation, an estimation equation of the squeezing parameter is given by

$$\tanh^2 \xi \approx \frac{2R}{1-2R} \quad (\text{S32})$$

on Page 10 of Supplementary Materials.

Comment 5: Regarding the bibliography, I believe it is well-informed. However, I would like to add the following two papers to enhance the references:

Paesani, S., Ding, Y., Santagati, R. et al. Generation and sampling of quantum states of light in a silicon chip. *Nat. Phys.* 15, 925–929 (2019). <https://doi.org/10.1038/s41567-019-0567-8>

<https://www.nature.com/articles/s41567-019-0567-8>

William R Clements et al 2018 *J. Phys. B: At. Mol. Opt. Phys.* 51 245503

DOI 10.1088/1361-6455/aaf031

<https://iopscience.iop.org/article/10.1088/1361-6455/aaf031>.

Answer 5: As suggested by the Reviewer, the two references are added to the revised manuscript as

“We further calculate the difference $C = F_Q - F_C$ between the fidelity of the reconstructed FC profile F_Q and the optimal fidelity obtainable from a classical strategy F_C to benchmark against a classical simulation [31, 32].” in Line 20 on Page 8.

Comment 6: *I have a few reservations about the lack of precision regarding certain terms related to quantum mechanics or optics in general.
' For example, terms like "squeezed photons" can not be used for referring to a squeezed vacuum state.*

Answer 6: As pointed out by the Reviewer, such a term is removed in the revised manuscript as “To increase the input light power for enlarging the squeezing level of the squeezed vacuum state, an Erbium-doped fiber amplifier (EDFA) is connected to amplify the laser power, and a pair of wavelength division multiplexer (WDM) is used to filter the pump signal.” in Line 23 on Page 6.

Proposed corrections:

Abstract :

47 - “Universal programmable 16-mode interferometer.” The universality of an interferometer is questionable

Answer: It is revised to “a fully programmable 16-mode interferometer” in Line 16 on Page 2.

Introduction :

150 - “Fork” → Fock

Answer: It is revised to “Fock basis” in Line 8 on Page 6.

150 - “ $\hat{R}(\tilde{U})$ ” → I would prefer the notation in the supplementary note for the rotation operator : \hat{R}_U (RU)

Answer: It is revised to “squeezing operations \hat{S}_r , rotation operations \hat{R}_U ” in Line 7 on Page 6.

151 - Notation : We would prefer using r as a squeezing parameter and not λ

Answer: It is revised to “The squeezing value r and unitary matrix U are given by the Takagi-Autonne decomposition of matrix A yields $A = U^T (\tanh(r)) U$.” in Line 9 on Page 6.

Quantum experimental framework :

166 - “Squeezing level of photons” gives the impression that the state is a squeezed Fock state. We would rather use “the squeezing parameter” or just the “Squeezing level of the squeezed vacuum state”.

Answer: It is revised to “the squeezing level of the squeezed vacuum state” in Line 23 on Page 6.

166 - Erbium dope fiber amplifier (EDFA) -> Erbium doped fiber amplifier (EDFA)

Answer: It is revised to “Erbium doped fiber amplifier” in Line 23 on Page 6.

Overall, this sentence is a little difficult to understand :

‘To increase the input light power for enlarging the squeezing level of photons, an Erbium doped fiber amplifier (EDFA) and a pair of Wavelength Division Multiplexer (WDM) are employed to amplify input light and suppress light power at wavelengths.’

Would it be possible to rephrase it ?

Answer: It is revised to “To increase the input light power for enlarging the squeezing level of the squeezed vacuum state, an Erbium doped fiber amplifier (EDFA) is connected to amplify the laser power, and a pair of wavelength division multiplexer (WDM) is used to filter the pump signal.” in Line 23 on Page 6.

168 - Rephrase the whole sentence to be clearer. “To amplify the light and select only the pump wavelength” for example.

Answer: It is revised to “To increase the input light power for enlarging the squeezing level of the squeezed vacuum state, an Erbium doped fiber amplifier (EDFA) is connected to amplify the laser power, and a pair of wavelength division multiplexer (WDM) is used to filter the pump signal.” in Line 23 on Page 6.

170 - The spiral lines are actually not visible on the figure.

Would it be possible to change the resolution of the picture? or place an inset with a zoom on the squeezer.

Answer: Figure 2 is revised with high resolution on Page 2 of the figure.

170 - Moreover, what is the role of the spiral in the vacuum squeezed state generation?

Is it a single pass squeezer or a cavity ring? Could the author be more specific?

Answer: It is revised to “The squeezing source is applied via a degenerate spontaneous four-wave mixing (SFWM) process from the spiral lines, which is a single-pass squeezer. The unitary matrix is then performed in the evolution through the MZI network.” in Line 28 on Page 6.

176 - “superconducting single-photon detectors.” → superconducting single-photon detectors (SNSPDs).

Answer: It is revised to “superconducting single-photon detectors (SNSPDs).” in Line 6 on Page 7.

213 - Is it possible to clarify how the author retrieved the fidelity?

Answer: It is revised to “A high fidelity F (inner product of two sequences divided by the product of their lengths) of 98.2% is achieved in this case” in Line 15 on Page 8.

Conclusion :

285 - Summarization - I did not understand. Did the author want to say summation?

Answer: It is revised to “summation” in Line 9 on Page 11.

SUPPLEMENTARY NOTE 1 :

_ number of photons p_j , we usually take n_j to avoid confusion with the probability notation.

Answer: p is a pattern. To avoid confusion, we changed p to n in SUPPLEMENTARY NOTE 1.

_ Eq S17: what is ζ ?

Answer: It is revised to “ ζ is the vector formed by $\zeta = \alpha - B\alpha^*$ ” in Line 22 on Page 4 of Supplementary Materials.

SUPPLEMENTARY NOTE 2 :

_ Refer to S2 figure but the figure is S1

Answer: It is revised to “The number of sums t in the sampling process vs. molecular size is shown in Figure S1 and the molecular with the size from 2 to 15 are simulated.” in Line 18 on Page 4 of Supplementary Materials.

SUPPLEMENTARY NOTE 3 :

_ “and a central wavelength of 1550.116nm and a bandwidth of 1.9nm” → “, a central wavelength of 1550.116nm and a bandwidth of 1.9nm”

Answer: It is revised to “, a central wavelength of 1550.116nm and a bandwidth of 1.9nm” in Line 11 on Page 7 of Supplementary Materials.

_ squeezing level of photons → Photons are not squeezed

Answer: It is revised to “To increase the input light power for increasing the squeezing level of a squeezed vacuum state, an Erbium dope fiber amplifier (EDFA) and another pair of WDM are employed to amplify the input light and suppress the light power at wavelengths other than the two selected ones, respectively.” in Line 14 on Page 7 of Supplementary Materials.

_squeezed photons

Answer: It is revised to “Up to sixteen pairs of single-mode squeezed vacuum state can be generated by the spontaneous four-wave mixing (SFWM) process from the sixteen spiral lines (a single-pass squeezer) on the integrated photonic chip.” in Line 19 on Page 7 of Supplementary Materials.

_ What are the spiral lines? Resonators? Why is it so important that it is a spiral line?

Answer: It is revised to “Up to sixteen pairs of single-mode squeezed vacuum state can be generated by the spontaneous four-wave mixing (SFWM) process from the sixteen spiral lines (a single-pass squeezer) on the integrated photonic chip.” in Line 19 on Page 7 of Supplementary Materials.

B. integrated photonic chip :

_ [0, 3]mI ?

Answer: It is revised to “[0, 3] mA” in Line 20 on Page 8 of Supplementary Materials.

_squeezing photons

Answer: It is revised to “For the source, the spiral line composed of single-mode waveguides (2 cm length) is used to produce a squeezed vacuum state” in Line 20 on Page 8 of Supplementary Materials.

C. Detectors and Probabilistic number resolving :

_ detectors’ → detector’s

Answer: It is revised to “detector’s” in Line 12 on Page 9 of Supplementary Materials.

_ I did not really see how did the author incorporate the efficiency in the probability formula

Answer: As pointed out by the Reviewer, the detector efficiencies are not included in the probability formula. Considering the efficiency of each detector is slightly different in our experiment, we normalized the detector's efficiency before the probability calculation.

SUPPLEMENTARY NOTE 4: Device performance :

A. Source characterization

_ By using the word "FSR" did the author mean Free Spectral Range? FSR is used for resonators not for MZI.

Answer: As pointed out by Reviewer, FSR is Free Spectral Range, which can be used for AMZI (asymmetric MZI). The related part is revised as

"To pump the on-chip sources, laser pulses are first generated and the AMZI with free spectral range (FSR) of 6.32 nm is used to filter the generated squeezed state from spiral lines." *in Line 11 on Page 10 of Supplementary Materials.*

_The filter is on the pump so it is not the generated squeezed state which is filtered or I may not understand.

Answer: The filter is used to remove the pump light from the chip, while retaining the generated squeezed state on the chip for further operations.

_degenerate SFWM it would be good if the author precise it in the main text

Answer: It is revised to "The squeezing source is applied via a degenerate spontaneous four-wave mixing (SFWM) process from the spiral lines, which is a single-pass squeezer. The unitary matrix is then performed in the evolution through the MZI network." *in Line 28 on Page 6.*

_(fig S3) Squeeze parameter → Squeezing parameter

Answer: It is revised to "Squeezing parameter" *in Line 28 on Page 6 of Supplementary Materials.*

_ Squeezing in dB ?

Answer: It is revised to "When one of the sources is connected directly to the detectors, the calculated squeeze value by second-order correlation measurements can reach around 0.3 (2.6 dB) when input power is around 0.38 mW (Figure S3a), and the count value of photons is around 4 kHz under this input power (Figure S3b)." *in Line 15 on Page 10 of Supplementary Materials.*

_ "When one of the sources is connected directly to the detectors, the calculated squeezing value by second-order correlation measurements can reach around 0.6 when input power is around 0.55mW (Figure S3a)" : Could the author explain how the squeezing parameter is linked to the second-order correlation ?

Answer: the measurements of squeezing from two-photon correlations are added in Supplementary Note 4 as

"The source emission's single-mode squeezing (SMS) parameter is calculated using two-mode second-order correlation measurements in the experiment setup schematized in Figure 2. Ref. [2] reported the calculation of the two-mode squeezing (TMS) parameter of the source using two-

mode second-order correlation measurements. Our method is similar to this by changing the source state to SMS state instead of TMS. Given a pure single-mode SMS state, the probability of detecting n_s (n_i) single photons in the signal (idler) mode in the loss-less case is expressed as

$$p_{SMS}(n_s, n_i) = \frac{\tanh^{2n_s}(\xi) (2n_s)!}{\cosh(\xi) 2^{2n_s} n_s! n_s!} \delta_{n_s, n_i} = p_0(n_s) \delta_{n_s, n_i} \quad (S25)$$

In the presence of losses, the detection probability is given by

$$\begin{aligned} p(n_s, n_i) &= \left(\frac{\eta}{1-\eta}\right)^{n_s+n_i} \sum_{n \geq \max\{n_s, n_i\}} p_0(n) (1-\eta)^{2n} \\ &= A(\xi, \eta) \left(\frac{\eta}{1-\eta}\right)^{n_s+n_i} \sum_{n \geq \max\{n_s, n_i\}} \frac{\tanh^{2n}(\xi) (2n)!}{2^{2n} n! n!} (1-\eta)^{2n} \\ &= A(\xi, \eta) \left(\frac{\eta}{1-\eta}\right)^{n_s+n_i} \sum_{n \geq \max\{n_s, n_i\}} \tanh^{2n}(\xi) \frac{1}{\sqrt{\pi n}} (1-\eta)^{2n} \\ &= A(\xi, \eta) \left(\frac{\eta}{1-\eta}\right)^{n_s+n_i} \sum_{n \geq \max\{n_s, n_i\}} (a_1 b_1^{2n} + a_2 b_2^{2n} + a_3 b_3^{2n}) \tanh^{2n}(\xi) (1-\eta)^{2n} \\ &= A(\xi, \eta) \left(\frac{\eta}{1-\eta}\right)^{n_s+n_i} \left\{ \begin{aligned} &a_1 \left(\sum_{n=0}^{\infty} [b_1 \tanh(\xi) (1-\eta)]^{2n} - \sum_{n=0}^{\max\{n_s, n_i\}} [b_1 \tanh(\xi) (1-\eta)]^{2n} \right) \\ &+ a_2 \left(\sum_{n=0}^{\infty} [b_2 \tanh(\xi) (1-\eta)]^{2n} - \sum_{n=0}^{\max\{n_s, n_i\}} [b_2 \tanh(\xi) (1-\eta)]^{2n} \right) \\ &+ a_3 \left(\sum_{n=0}^{\infty} [b_3 \tanh(\xi) (1-\eta)]^{2n} - \sum_{n=0}^{\max\{n_s, n_i\}} [b_3 \tanh(\xi) (1-\eta)]^{2n} \right) \end{aligned} \right\} \\ &= B(\xi, \eta) \left(\frac{\eta}{1-\eta}\right)^{n_s+n_i} \left\{ \frac{a_1}{q_1(\xi, \eta)} [b_1 \tanh(\xi) (1-\eta)]^{2\max\{n_s, n_i\}} \right. \\ &\quad \left. + \frac{a_2}{q_2(\xi, \eta)} [b_2 \tanh(\xi) (1-\eta)]^{2\max\{n_s, n_i\}} + \frac{a_3}{q_3(\xi, \eta)} [b_3 \tanh(\xi) (1-\eta)]^{2\max\{n_s, n_i\}} \right\} \end{aligned} \quad (S26)$$

where $\{a_k, b_k\} = \{0.5913, 0.4883\}; \{0.2705, 0.9243\}; \{0.1357, 0.995\}$, $q_i = 1 - b_i \tanh(\xi) (1-\eta)$ ($i = 1, 2, 3$) and $A(\xi, \eta)$ and $B(\xi, \eta)$ are normalization constants.

To determine the constant $B(\xi, \eta)$, the normalization condition $\sum_{n_s, n_i=0}^{\lim} p(n_s, n_i) = 1$ is imposed and rewrite as

$$\sum_{n_s, n_i=0}^{\lim} p(n_s, n_i) = 1 = 2 \sum_{n_i=0}^{\lim} \sum_{n_s \geq n_i} p(n_s, n_i) - \sum_{n_i=0}^{\lim} p(n, n) \quad (S27)$$

and

$$\frac{a_1 B(\xi, \eta)}{\xi(a) (1-\eta^2 b_1^2 \tanh^2 \xi) q_1(\xi, \eta)} + \frac{a_2 B(\xi, \eta)}{\xi(b) (1-\eta^2 b_2^2 \tanh^2 \xi) q_2(\xi, \eta)} + \frac{a_3 B(\xi, \eta)}{\xi(c) (1-\eta^2 b_3^2 \tanh^2 \xi) q_3(\xi, \eta)} = 1 \quad (S28)$$

where $a = \eta(1-\eta) \tanh^2 \xi b_1^2$, $b = \eta(1-\eta) \tanh^2 \xi b_2^2$, $c = \eta(1-\eta) \tanh^2 \xi b_3^2$ and $\xi(x) = (1-x)/(1+x)$.

Then, correlation measurements at time delay $\Delta t = 0$ thus represent events whenever $n_s \geq 1$ signal photons and $n_i \geq 1$ idler photons from a single SMS arrive at the detectors simultaneously, which happens with a probability

$$\begin{aligned}
p_{\text{coinc}}(\Delta t = 0) &= p(n_s \geq 1, n_i \geq 1) & (S29) \\
&= 1 - p(0, 0) - p(0, 1) - p(1, 0) \\
&= 1 - \left(\frac{a_1}{q_1} + \frac{a_2}{q_2} + \frac{a_3}{q_3} \right) B - 2 \left(\frac{a_1 b_1^2}{q_1} + \frac{a_2 b_2^2}{q_2} + \frac{a_3 b_3^2}{q_3} \right) B \tanh^2(\xi) \eta (1 - \eta)
\end{aligned}$$

The probability of having this event with a time delay $\Delta t > 0$ is

$$\begin{aligned}
p_{\text{coinc}}(\Delta t > 0) &= p(n_s \geq 1, n_i \geq 1) & (S30) \\
&= \left(\sum_{n_i=0}^{\text{lim}} p(n_s \geq 1, n_i) \right) \left(\sum_{n_s=0}^{\text{lim}} p(n_s, n_i \geq 1) \right) \\
&= \left(1 - \frac{a_1 B}{q_1(1-a)} - \frac{a_2 B}{q_2(1-b)} - \frac{a_3 B}{q_3(1-c)} \right)^2
\end{aligned}$$

Then, the quantity to be considered is the ratio between the coincidences measured at $\Delta t > 0$ and those measured at $\Delta t = 0$,

$$R = \frac{p_{\text{coinc}}(\Delta t > 0)}{p_{\text{coinc}}(\Delta t = 0)} \quad (S31)$$

In our experiment, the squeezing parameters are set small and losses in the detection channels are significant. Under this approximation, an estimation equation of the squeezing parameter is given by

$$\tanh^2 \xi \approx \frac{2R}{1 - 2R} \quad (S32)''$$

on Page 11.

C. Quantum characterization

– “and they are always higher than 0.9 for two-photon clicks and higher than 0.8 for four-photon clicks.” - If the fidelities are given in percent in the previous sentence, this fidelities could also be in percent.

Answer: It is revised to “The measured average fidelity in our experiment of two-photon and four-photon clicks in the output is shown in Figure S3c and they are always higher than **90%** for two-photon clicks and higher than **80%** for four-photon clicks.” in Line 15 on Page 14 of *Supplementary Materials*.

– It would be good to compare fidelities obtained using the quantum circuit with fidelities obtained using a classical algorithm.

Answer: It is revised to “**A small improvement over a classical strategy ($C = 1.7\%$) is obtained for formic acid molecules.**” in Line 25 on Page 8.

SUPPLEMENTARY NOTE 6: Data encoding details

– What are r_1, r_2 ? Squeezing parameters?

Answer: These parameters are changed to ω_1 and ω_2 , which are the real constant rescale weights.

SUPPLEMENTARY NOTE 7: Error and scalability analysis

– Why did the author place a polarization controller before the detector?

Answer: Since the detector is a polarization-sensitive device, a polarization controller is added before the detector. This is discussed as

“Finally, considering the detector is a polarization-sensitive device, the off-chip filters and the polarization controller are added before the detectors. The measured average transmission efficiencies were found to be 0.6 dB and 0.1 dB, respectively, and averaged over the 16 channels.” *in Line 23 on Page 18 of Supplementary Materials.*

SUPPLEMENTARY NOTE 8: Derivation of Non-Condon effects

_no-Condon → non-Condon

Answer: It is revised to “non-Condon” *in Line 14 on Page 19 of Supplementary Materials.*

_ (3), they propose → Cite the authors

Answer: It is revised to “Huh et al. propose a theoretical instruction for simulating the non-Condon spectrum for the linear and quadratic HT terms with a GBS approach [4].” *in Line 17 on Page 19 of Supplementary Materials.*

In summary, all reviewer comments are addressed. In addition, the manuscript and supplementary materials are carefully proofread.

REVIEWER COMMENTS

Reviewer #1 (Remarks to the Author):

The revised version of the paper by Zhu et al. shows some improvements compared to the previous version, but significantly more improvement is needed for it to be published. The current version is not acceptable.

As I wrote in the previous report, I value the new idea in the paper. It uses a new loop-Hafnian expansion theory in terms of Hafnian for the analog simulation of molecular vibronic spectroscopy without invoking displacements. Unfortunately, it still needs much improvement. Even glancing, I can see errors, misconceptions, and incorrect terminologies. The writing has to be seriously improved to be published elsewhere. It definitely needs language correction services. I don't list all of them, but I leave some comments on the manuscript for improvement.

1. In the abstract: "Handling experimentally...an open problem", I don't understand this sentence.
2. In the abstract: In stating the complexity of the problem, please carefully state.
3. In the abstract: "linear network" -> "linear optical network"
4. In the abstract, I advise the authors to write what they have done. The complexity of the molecular problem requires further work.
5. line 66: "chemical reaction mechanistic information", what is this?
6. Lines 66-67: "force-field changes, vibrational excitations, electron transport" should not be listed together as different applications.
7. line 71: "inverse problem", what is this?
8. First paragraph in the introduction: Consider rewriting it totally with more citations. The logical flow is odd.
9. I often find the misconception about the difference between photoelectron spectroscopy and optical spectroscopy.
10. Lines 92-93: The author's main claim is the squeezed coherent states are not needed in the photonic simulation. I think it is important too. However, other physical platforms, like trapped ions and superconducting devices, do not have the limitations of photonic systems.

Compare the photonic system to other platforms to discuss this and justify why the photonic system needs this technique. Why photonics?

11. Lines 103-104: Normally, "squeezed vacuum" is used instead of "vacuum squeezed", and is change "linear network" to "linear optical network". The sentence is odd. Does the squeezed vacuum make the FCF calculation easy? This is only for the experimental implementation.

12. The molecular problem is an interesting application but is still far from practical. Comment on how your technique can be useful in other applications. It is not necessarily a physical problem, but it can be a technical problem in quantum optics.

13. line 119: "relative vibrational wave functions", strange to me.

14. lines 122-123: the terms in the sentence are not equal, surface is not an oscillator, for example. I often see such sentences in the manuscript.

15. line 136: "the vibronic spectroscopy of the molecule", this should refers to the theoretical values but it sounds like the real experimental spectroscopic values.

16. Eq.1: Comment on the loop-hafnian. FCF is loop-hafnian. Present how it is calculated conventionally, and then present your expression for the comparison. The paper should be presented in a better way that focuses on the author's method. Problem->Motivation->Solution, I think the application should only be used as a demonstration and suggest not to focus on the application itself. It has been there for a long time.

17. line 144-145: rewrite the last clause, it sounds strange.

18. line 148: in the equation, sigma is not defined. I strongly suggest authors recheck all the terms used.

19. line 154: cite Takagi-Autonne decomposition

20. line 230-231: What improvement compared to what?

21. line 267, comment on the fidelities, 86.3 % does not look good, why?

22. line 270: "The molecular parameters are concluded..", sounds strange, rewrite it.

23. In the conclusion, the authors claim they simulated anharmonicity. It is not true; all the molecular problems in the manuscript are in harmonic picture, including the non-Condon part.

24. line 322: Not only in this line but throughout the paper, when the authors comment on the complexity, make it clearer and be careful in stating about it.

25. line 377: non-Condon anharmonic effect -> non-Condon effect

26. line 362-363: The sentence is not understandable.
27. Please check variables in italic or in roman. Equations should end with punctuation or comma.
28. In SM: delta is used for two different purposes.
29. In SM (Eq. S21-S23): Still, needs more work. It contains errors or typos, B_{ij} . They can be presented better. I think it is not well presented in a logical way. Improve the derivation by focusing on how to tweak the diagonal part out of A and converting the loop hafnian in terms of hafnians so that only the squeezed vacuums are needed in the experiments.
30. "combinatorial number"-> "binomial coefficient"
31. Eq. S24: The authors must prove the last equation. The equality works only when the Hafnian is positivesemidefinite. The authors should prove this. Otherwise, the equation is only sometimes acceptable; it would only work for some special cases. Eq. 2 should appear in the main text with an explanation.

Reviewer #2 (Remarks to the Author):

The authors have significantly re-written the manuscript and have addressed many of my concerns. Unfortunately, I believe that there are still some significant omissions to be corrected and clarifications to be added prior to publication.

The most glaring omission is the source of Franck-Condon and Herzberg-Teller parameters. These parameters are not inherent properties of a molecule. Rather, they are either derived from electronic structure calculations at a certain level of theory, or they are fit to a molecular spectrum. The former depends on the level of theory employed, and the latter depends on the experimental conditions and resolution. The authors must provide a source for all parameters in Supplementary Note 5. If the authors calculated the parameters themselves, the program and level of theory should be cited. If the parameters come from a previous study, using them without citation is plagiarism.

The authors have changed their formic acid example to only 4 modes. However, their U matrix is now non-unitary. How did the authors implement a non-unitary transformation in the experiment?

In the sentence "... the vibronic spectroscopies of two molecules with a point-group symmetry are successfully simulated experimentally on the quantum photonic chip", I fail to see the relevance of the symmetries. Both molecules belong to the C_s point group, which only has two irreducible representations. The reduction in the number of Franck-Condon active modes is only partially explained by symmetry.

The performance of experiments is very difficult to assess in Fig. 4, especially subfigure c. The authors added that the benzene experiment has a fidelity of 99.4%, but the grey bar for the experiment is only visible on a single peak.

In the section "Non-Condon effects in molecular spectroscopy", the authors appear to have mixed up references 33 and 34.

Reviewer #3 (Remarks to the Author):

In this revised paper, the author diligently explained, and corrected all the points raised in the first review. I do not see any major flaws with the experimental setup and the quantum aspects, apart from some imprecise terms that give the impression that the authors sometimes do not have a strong grasp of the subject matter..

For example : In the supplementary notes, it is more appropriate to use the term "squeezing parameter", or "squeezing value", rather than "squeeze parameter" or "squeeze value".

Despite this clever idea of doubling the number of squeezed vacuum modes rather than using squeezed coherent states, and the demonstration that this method indeed has a quantum advantage, this advantage seems not as significant compared to the computational resources that the authors need to employ. This seems likely due to the low squeezing parameter of the source and losses.

Based on the results, I personally have the overall feeling that this article could be a good fit for another journal or a different section of this publication.

Answer 1 :

I do not agree with the first author's argument which explains that the splitting ratio is difficult to set precisely for the squeezed state and the coherent state interference. Indeed, it seems that regarding the programmable interferometer, the difficulty in setting the splitting ratio of the beam splitter of this interferometer should be the same in this case.

I however agree with the second argument concerning scalability. The phase locking and the coherent state modulation requires indeed more bulk devices.

To summarize, I globally agree with the proposed correction.

Answer 2: The addition of the fidelity of naphthalene, phenanthrene and benzene under the non-Condon regime makes this article more precise and clear.

Answer 3 : The chip shows a small quantum advantage compared to classical resources (1.9% for FC and 1.3 % for non-Condon).

It would be good if the author can explain why the advantage is low and how it could be improved.

Answer 4: I appreciate the helpful and detailed explanation. Here are typo errors in the supplementary note :

p.10 squeeze value → squeezing value

p.11 the limit of the sum in the normalization condition.

p.11 eq S26 Unclosed brace '{'

All the other proposed corrections were implemented. I however see other typo errors:

Supplementary note I :

p.3 There is an open brace $f(g(t))$

$f(g(t)) \rightarrow f(g(t))$

p.7 the complicity → the complexity ?

Manuscript ID: Nature Communications manuscript NCOMMS-23-34789A

Paper title: Large-scale photonic network with squeezed vacuum states for molecular vibronic spectroscopy

Authors: H. H. Zhu, H. S. Chen, T. Chen, Y. Li, S. B. Luo, M. F. Karim, X. S. Luo, F. Gao, Q. Li, H. Cai, L. K. Chin, L. C. Kwek, B. Nordén, X. D. Zhang, and A. Q. Liu

Reply to Reviewer 1

We are grateful to the Reviewer for constructive comments and recognition of our experimental achievements and contributions. We appreciate the valuable suggestions, which have catalyzed great improvements to our manuscript. We are happy to address all issues.

Comment 1: *The revised version of the paper by Zhu et al. shows some improvements compared to the previous version, but significantly more improvement is needed for it to be published. The current version is not acceptable.*

As I wrote in the previous report, I value the new idea in the paper. It uses a new loop-Hafnian expansion theory in terms of Hafnian for the analog simulation of molecular vibronic spectroscopy without invoking displacements. Unfortunately, it still needs much improvement. Even glancing, I can see errors, misconceptions, and incorrect terminologies. The writing has to be seriously improved to be published elsewhere. It definitely needs language correction services. I don't list all of them, but I leave some comments on the manuscript for improvement.

Answer 1: As suggested by the Reviewer, we have made a significant revision to the manuscript, making it more legible and comprehensive.

Comment 2: *In the abstract: "Handling experimentally...an open problem", I don't understand this sentence.*

Answer 2: As pointed out by the Reviewer, the statement is revised in the **Abstract** as

“Quantum simulation, though theoretically promising, faces technological challenges in experimentally extracting vibronic spectra for molecules with multiple modes.” on Page 2.

Comment 3: *In the abstract: In stating the complexity of the problem, please carefully state.*

Answer 3: As suggested by the Reviewer, we have changed the corresponding content to a detailed explanation of complexity in **Discussion** (see also **Answer 24**) as

“Our approach enables the generation of molecular vibronic spectra through the modified boson sampling in an optical network. Compared with the traditional boson sampling, our algorithm incorporates an additional weighted summation of the hafnian function. It is noted that the number of summation terms is smaller than the sampling number. As a result, the scalability of the

algorithm mainly depends on the sampling process (see details in **Methods** and **Supplementary Note 2**).” on Page 11.

Comment 4: *In the abstract: "linear network"->"linear optical network".*

Answer 4: As pointed out by the Reviewer, we have revised the sentence as

“Here, we propose a nontrivial algorithm to generate the vibronic spectra using states with zero displacements (squeezed vacuum states) coupled to a linear optical network, offering ease of experimental implementation.” on Page 2.

Comment 5: *In the abstract, I advise the authors to write what they have done. The complexity of the molecular problem requires further work.*

Answer 5: As suggested by the Reviewer, we have revised the **Abstract** as

“Although molecular vibronic spectra generation is pivotal for chemical analysis, tackling such exponentially complex tasks on classical computers remains inefficient. Quantum simulation, though theoretically promising, faces technological challenges in experimentally extracting vibronic spectra for molecules with multiple modes. Here, we propose a nontrivial algorithm to generate the vibronic spectra using states with zero displacements (squeezed vacuum states) coupled to a linear optical network, offering ease of experimental implementation. We also fabricate an integrated quantum photonic microprocessor chip as a versatile simulation platform containing 16 modes of single-mode squeezed vacuum states and a fully programmable interferometer network. Molecular vibronic spectra of formic acid and thymine under the Condon approximation are simulated using the quantum microprocessor chip with high reconstructed fidelity (> 92%). Furthermore, vibronic spectra of naphthalene, phenanthrene, and benzene under the non-Condon approximation are also experimentally simulated. Such demonstrations could pave the way for solving complicated quantum chemistry problems involving vibronic spectra and computational tasks beyond the reach of classical computers.” on Page 2.

Comment 6: *line 66: "chemical reaction mechanistic information", what is this?*

Answer 6: As pointed out by the Reviewer, we deleted the inappropriate text.

Comment 7: *Lines 66-67: "force-field changes, vibrational excitations, electron transport" should not be listed together as different applications.*

Answer 7: As pointed out by the Reviewer, we have rewritten the applications of molecular vibronic spectra as

“Exploration of molecular vibronic spectra is vital for understanding molecular properties in chemical analysis and biological labels [1-7].” on Page 3.

Comment 8: *line 71: "inverse problem", what is this?*

Answer 8: As pointed out by the Reviewer, the term “inverse problem” has been deleted in the revised manuscript.

Comment 9: *First paragraph in the introduction: Consider rewriting it totally with more citations. The logical flow is odd.*

Answer 9: As suggested by the Reviewer, we have rewritten the first paragraph of the **Introduction** as

“Exploration of molecular vibronic spectra is vital for understanding molecular properties in chemical analysis and biological labels [1-7]. Vibronic spectroscopy typically involves the characterization of molecular vibronic transitions, which are simultaneous changes in the electronic and vibrational energy levels of a molecule [8]. The resulting vibronic transition probabilities between two electronic states are proportional to the Franck-Condon factors (FCF) [8-10]. Given its pivotal role in chemistry, various strategies on classical computers have been extensively studied. Among them, one renowned algorithm is the eigenvalue-trace algorithm, in which the running time scales exponentially in the system size [11, 12]. Thus, it remains a long-standing computationally difficult problem that cannot be efficiently solved using classical computers. Recently, the advent of quantum simulation [13-16], a groundbreaking development at the intersection of computer science and quantum mechanics, holds the promise of overcoming the computational challenges associated with exponential computing time, thereby opening up new avenues for advanced molecular studies.” *on Page 3.*

Comment 10: *I often find the misconception about the difference between photoelectron spectroscopy and optical spectroscopy.*

Answer 10: As pointed out by the Reviewer, the **photoelectron spectroscopy** is an experimental technique used to determine the relative energies of electrons in atoms and molecules. **Optical spectroscopy** is a technique used to study the material properties and molecular structure of the sample. **Vibronic spectroscopy** is a branch of molecular spectroscopy concerned with vibronic transitions. In our manuscript, we focus on molecular vibronic spectroscopy and all related statements have been corrected to the molecular vibronic spectroscopy as

“Vibronic spectroscopy typically involves the characterization of molecular vibronic transitions, which are simultaneous changes in the electronic and vibrational energy levels of a molecule [8]. The resulting vibronic transition probabilities between two electronic states are proportional to the Franck-Condon factors (FCF) [8-10].” *on Page 3.*

“Molecular vibronic spectra describe the molecular vibronic transition, and the vibronic spectral profile, called the Franck-Condon profile (FCP), is obtained by computing the corresponding FCF, which is the square of the overlap integral of two wave functions in different electronic levels within the Born-Oppenheimer approximation and the Condon approximation [31-33].” *on Page 4.*

“In contrast, when considering the linear HT operator, the vibronic spectral progression is obtained (see red bars located above the x-axis in **Fig. 4a**).” *on Page 10.*

Comment 11: *Lines 92-93: The author's main claim is the squeezed coherent states are not needed in the photonic simulation. I think it is important too. However, other physical platforms, like trapped ions and superconducting devices, do not have the limitations of photonic systems. Compare the photonic system to other platforms to discuss this and justify why the photonic system needs this technique. Why photonics?*

Answer 11: As pointed out by the Reviewer, the implementation of squeezed coherent states indeed has fewer limitations in trapped ions and superconducting devices. However, most superconducting computers are **too susceptible to noise**, have low computational fidelities and require a **low working temperature**, which limits the simulation of large molecules. For trapped-ion technology, as the number of ions increases, **manipulating** the motional states of ions is **harder** with a **low working temperature**. Many ions are also plagued by gate error sources like anomalous heating from the surface effects of electrodes and crosstalk. On the other hand, photonic systems operate **at room temperature** and are generally **less susceptible to lossy errors**.

A detailed discussion is added as

“Quantum algorithms for the simulation of molecular vibronic spectra have been demonstrated in systems with superconducting qubits [17] and trapped ions [18, 19], such as the experimental implementation of a boson sampling protocol and construction of the vibrational Hamiltonian in a standard quantum circuit. Yet, the study of molecules with multiple modes is hampered by the limited gate fidelities or the susceptibility to various noise sources in superconducting or trapped-ion systems [20, 21]. In addition, the low working temperature requirements in the two systems also pose challenges for system miniaturization [22, 23]. As a result, only molecules with two vibronic modes have been achieved experimentally. Meanwhile, the advantages of photons, such as their ease of manipulation, precise modulation, and capacity to operate at room temperature, underscore their potential for photonic systems as a promising platform for molecular simulation [24].” on Page 3.

Comment 12: *Lines 103-104: Normally, "squeezed vacuum" is used instead of "vacuum squeezed", and is change "linear network" to "linear optical network". The sentence is odd. Does the squeezed vacuum make the FCF calculation easy? This is only for the experimental implementation.*

Answer 12: As pointed out by the Reviewer, all terms have been rectified: “squeezed vacuum”, “linear optical network” and “squeezed coherent states”. In addition, as suggested by the Reviewer, the model with squeezed vacuum sources and a linear optical network is used to calculate the FCF. In contrast to **quantum algorithms using squeezed coherent states**, our approach simplifies experimental implementation, enabling the successful realization of large-scale molecular vibronic simulations. Meanwhile, it has a similar computational simplicity on algorithm runtime as quantum algorithms employing squeezed coherent states.

In the revised manuscript, the advantages of our approach are discussed in the **Abstract** as

“Here, we propose a nontrivial algorithm to generate the vibronic spectra using states with zero displacements (squeezed vacuum states) coupled to a linear optical network, offering ease of experimental implementation.” on Page 2.

Comment 13: *The molecular problem is an interesting application but is still far from practical. Comment on how your technique can be useful in other applications. It is not necessarily a physical problem, but it can be a technical problem in quantum optics.*

Answer 13: As pointed out by the Reviewer, the quantum photonic circuit has many practical applications beyond the vibronic spectra of molecules, which is discussed in the revised manuscript as

“Our scheme and integrated quantum photonic chip open new avenues for numerous practical applications in quantum chemistry, such as molecular docking problems [29] and quantum machine learning, including graph classification [30].” on Page 4.

Comment 14: *line 119: "relative vibrational wave functions", strange to me. lines 122-123: the terms in the sentence are not equal, surface is not an oscillator, for example. I often see such sentences in the manuscript.*

Answer 14: As suggested by the Reviewer, we have thoroughly revised the manuscript and corrected those ambiguous terms. For instance,

“Molecular vibronic spectra describe the molecular vibronic transition, and the vibronic spectral profile, called the Franck-Condon profile (FCP), is obtained by computing the corresponding FCF, which is the square of the overlap integral of two wave functions in different electronic levels within the Born-Oppenheimer approximation and the Condon approximation [31-33].” on Page 4.

“For a molecule with n vibrational modes, as shown in **Fig. 1a**, the quantum computing of the vibronic spectra needs to consider the transformation between two potential energy surfaces. Furthermore, the relationship between the initial and final energy surface described by the Doktorov transformation [31] can be expressed in terms of three quantum operations: squeezing, rotation, and displacement.” on Page 5.

Comment 15: *line 136: "the vibronic spectroscopy of the molecule", this should refer to the theoretical values but it sounds like the real experimental spectroscopic values.*

Answer 15: As suggested by the Reviewer, the vibronic spectroscopy of the molecule refers to the theoretical values. The theoretical part has been rewritten and the sentence has been deleted in the revised manuscript.

Comment 16: *Eq.1: Comment on the loop-hafnian. FCF is loop-hafnian. Present how it is calculated conventionally, and then present your expression for the comparison. The paper should be presented in a better way that focuses on the author's method. Problem->Motivation->Solution, I think the application should only be used as a demonstration and suggest not to focus on the application itself. It has been there for a long time.*

Answer 16: As suggested by the Reviewer, we would like to highlight some core points of our algorithm.

(1) The **problem** of calculating the FCF involves a well-known classical algorithm, specifically the eigenvalue-trace algorithm, which exhibits a complexity of $O(N^3 2^{N/2})$ with N being the matrix size. This implies that computing a single line on the Franck-Condon profile requires **exponential time complexity**. Moreover, obtaining the entire vibronic spectra demands a sheer number of outcomes exponentially, making the molecular vibronic spectra problem computationally hard (#P-hard).

(2) Our expression of FCF aligns with the loop-hafnian expression. The **motivation** for calculating FCF using the squeezed vacuum state is to **simplify experimental implementation**.

To better express our idea, we have rewritten the “*A squeezed vacuum state and linear optical network for approximated molecular vibronic spectra.*” section in the revised manuscript as

“**A squeezed vacuum state and a linear optical network for molecular vibronic spectra approximation.** Molecular vibronic spectra describe the molecular vibronic transition, and the vibronic spectral profile, called the Franck-Condon profile (FCP), is obtained by computing the corresponding FCF, which is the square of the overlap integral of two wave functions in different electronic levels within the Born-Oppenheimer approximation and the Condon approximation [31-33]. In classical computing, the FCF of a given vibronic molecular transition is equivalent to the loop hafnian of a particular matrix [31], which is a #P-hard task [6]. The development of quantum computing puts forward a new way, which leads to an efficient solution to the complicated calculation of FCF in the quantum frame. For a molecule with n vibrational modes, as shown in **Fig. 1a**, the quantum computing of the vibronic spectra needs to consider the transformation between two potential energy surfaces. Furthermore, the relationship between the initial and final energy surface described by the Doktorov transformation [31] can be expressed in terms of three quantum operations: squeezing, rotation, and displacement. As shown in **Fig. 1b**, the previous quantum computation for FCF requires producing squeezed coherent states with m modes. However, the experimental obstacle in squeezed coherent state preparation [27] limits the application of this method.

To perform the quantum computation of FCF effectively in the experiment, we build a relation between the loop hafnian (employed in calculating molecular vibronic spectra) and the hafnian. Since the boson sampling protocol establishes a link between the hafnian matrix function and the output pattern within easily attainable linear optics [34-36], it is straightforward to implement the FCF experimentally within the quantum frame. Through the mode expansion, the original displacement parameter of the i -th vibrational mode is represented by the covariance between the i -th and $(m+i)$ -th optical modes (see detailed derivation in **Supplementary Note 1**). The original vibrational transition with m modes is related to squeezing and rotation operations with $2m$ optical modes, as shown in **Fig. 1c.**” on Page 4.

Comment 17: line 144-145: rewrite the last clause, it sounds strange.

Answer 17: As pointed out by the Reviewer, the sentence has been revised as

“ $A^{(n,l)}$ is a submatrix of A depending on the measured output pattern (\mathbf{n}, \mathbf{l}) .” on Page 5.

Comment 18: line 148: in the equation, sigma is not defined. I strongly suggest authors recheck all the terms used.

Answer 18: As suggested by the Reviewer, we have thoroughly rechecked all the used terms and added the definition of sigma as

“Meanwhile, the probability amplitudes of an output pattern are proportional to the square of the hafnian function as $Pr(\mathbf{n}, \mathbf{l}) = \frac{1}{\mathbf{n}!! \sqrt{\sigma_Q}} |haf(A^{(\mathbf{n}, \mathbf{l})})|^2$, where $\sigma_Q = \sigma + I_{4m}/2$ with I_{4m} being the $4m \times 4m$ identity matrix and σ is the covariance matrix [36, 37].” on Page 6.

Comment 19: line 154: cite Takagi-Autonne decomposition.

Answer 19: As suggested by the Reviewer, Takagi-Autonne decomposition has been cited in Ref. [38] as

“The squeezing value \mathbf{r} and unitary matrix V are given by the Takagi-Autonne decomposition [38] of matrix A , resulting in $A = V^T(\tanh(\mathbf{r}))V$.” on Page 6.

Comment 20: line 230-231: What improvement compared to what?

Answer 20: As suggested by the Reviewer, we use the criteria: $C = F_Q - F_C$, the difference between the fidelity of the reconstructed Franck-Condon profile F_Q and the optimal fidelity obtainable from a classical strategy F_C to benchmark the quantum performance against a classical simulation.

The sentence has been revised as

“We further calculate the difference between the fidelity of the reconstructed FC profile F_Q and the optimal fidelity obtainable from a classical strategy F_C ($C = F_Q - F_C$) to benchmark against the classical simulation [41, 44]. The result shows that an improvement of FCP fidelity to the ideal FCP over the classical strategy ($C = 6.8\%$) is obtained for formic acid molecules.” on Page 9.

Comment 21: line 267, comment on the fidelities, 86.3 % does not look good, why?

Answer 21: As pointed out by the Reviewer, the fidelity of phenanthrene is 86.3%. We rechecked the calculation of fidelity and found out the mistake of data misalignment. We have corrected the error and obtained the fidelity of 98.9%. In the fidelity calculation, the used equation is $F = \frac{\sum_i p_i q_i}{|\mathbf{p}| |\mathbf{q}|}$, where $\{p_i\}$ and $\{q_i\}$ are normalized probability distributions from experiment and theory, yielding fidelities of 99.4%, 98.9%, and 99.4% for naphthalene, phenanthrene, and benzene, respectively. In addition, utilizing the equation $F_2 = \sum_i \sqrt{p_i q_i}$ from Ref. [41], we obtain fidelities of 98.4%, 98.4%, and 93.4% for naphthalene, phenanthrene, and benzene, respectively. Both methods confirm a high fidelity between our experiment and theory. Considering \mathbf{p} and \mathbf{q} are probability distributions, where $\sqrt{p_i}$ and $\sqrt{q_i}$ are related to probability amplitude, the inner product between vectors with $\sqrt{p_i}$ and $\sqrt{q_i}$ from Ref. [41] aligns with the definition of statistical overlap [42] and

is more widely accepted equation for this kind of fidelity calculation. Thus, we have modified the fidelity calculation equation to $F_2 = \sum_i \sqrt{p_i q_i}$.

The issue of fidelity is revised as

“The similarity between the experimentally reconstructed and theoretical FCPs is characterized by computing the fidelity of two sequences, $F = \sum_i \sqrt{p_i q_i}$, where $\{p\}$ and $\{q\}$ are the normalized theoretical and experimental probability distributions, respectively [41-43]. A fidelity F of 92.9% is achieved in this case, which is limited by the inevitable flaws in circuit fabrication and operation, photon noise, and photon loss.” on Page 8.

“The measured fidelities with the vibronic spectra of naphthalene, phenanthrene, and benzene are 98.4%, 98.4%, and 93.4%, respectively.” on Page 10.

Comment 22: line 270: “The molecular parameters are concluded..”, sounds strange, rewrite it.

Answer 22: As suggested by the Reviewer, we have rewritten the sentence as

“The molecular characteristic parameters, including vibrational frequencies and transformation matrices, are summarized in Supplementary Note 5.” on Page 10.

Comment 23: In the conclusion, the authors claim they simulated anharmonicity. It is not true; all the molecular problems in the manuscript are in harmonic picture, including the non-Condon part.

Answer 23: As suggested by the Reviewer, we have corrected the claim as

“To perform molecular simulation within the harmonic model, we demonstrate an on-chip reconfigurable quantum simulator achieved through an integrated quantum photonic microprocessor chip with 16 squeezed modes injected into an interferometer network.” on Page 11.

Comment 24: line 322: Not only in this line but throughout the paper, when the authors comment on the complexity, make it clearer and be careful in stating about it.

Answer 24: As suggested by the Reviewer, the analysis of the complexity of this problem has been discussed in detail. Simulating vibronic spectra is a task of exponential complexity that proves challenging for classical computers. Different from classical algorithms, we first build the connection between loop hafnian (mapped to the calculation of FCFs) and the summation of hafnian. Then, a linear optics device with an initial squeezing state can be used to test this simulation. To compete with the classical algorithm, one has to demonstrate the scalability of the quantum algorithm based on the experimental platform. Thus, we have added a more detailed analysis of the complexity of the algorithm in several sections.

(1) In Supplementary Note 2, we have discussed the complexity of the classical algorithm, quantum simulation based on squeezed coherent states, and our quantum algorithm.

“In this section, we discuss the complexity of obtaining molecular vibronic spectra in three aspects: classical computation, quantum simulation based on squeezed coherent states, and our

new method based on squeezed vacuum states. Calculating the molecular vibronic spectra with classical algorithms seems challenging due to two reasons. (a) The FCF associated with a transition between initial and final vibrational states in two different potential energy surfaces is equivalent to calculating the number of perfect matchings of a weighted graph with loops [6], which is known as the loop hafnian. Computing the loop hafnian is #P-hard, and the renowned classical algorithm, specifically the eigenvalue-trace algorithm, exhibits a complexity of $O(N^3 2^{(N/2)})$ with N being the matrix size [12]. (b) The density of vibrational states increases dramatically with the number of atoms and internal energy. Therefore, obtaining the entire FCP requires a large number of results, making the molecular vibration spectrum problem exponentially larger in molecular size.

The relationship between the Doktorov operator describing the transformation of vibronic state and the operators associated with displacement, squeezing and rotation in quantum optics is tactfully established, proposing a modification of boson sampling for the purpose of quantum simulation [10]. This approach enables us to convert this computationally challenging problem into a sampling task, allowing for the straightforward estimation of spectra at each given vibrational transition frequency (ω_{vib}) by collecting output patterns. We need to declare that the number of samples required to reach a desired precision (ϵ) scales as $O(\epsilon^{-2})$. It means that a polynomial number of samples is sufficient to estimate the FCP if the goal is to construct an overall profile with reasonable precision [10].

We construct the relationship between the FCF with a weighted summation of the hafnian function which corresponds to the probability distribution of a boson sampling problem with a squeezed vacuum state in **Supplementary Note 1**. In addition, doubling the original modes ensures that each sample maps to a single transition frequency and that the number of post-summation will not be greater than the number of samples. The detailed derivation of the scalability of the sampling process is shown in **Methods**. Furthermore, the numerical simulation is used to evaluate the number of summation terms. The relationship between the number of summation terms, denoted as t , and the molecular size is shown in **Fig. S3**, where the molecular size ranging from 2 to 15 are simulated. It can be seen that the probability of occurrence of many combinations of samples is extremely low, the average number of summation terms is lower than 2 and the maximum t does not increase with the molecular size. Thus, the sum operation does not add a lot of computational burdens. In **Fig. S3**, the fidelity between the theory and simulation versus molecular size with 10000 samplings is calculated. It is further verified from the simulation aspect that our method is sufficient to estimate the FCP of molecules.

In summary, when addressing the challenging task of simulating molecular vibronic spectra, the eigenvalue-trace algorithm exhibits an exponential runtime in terms of the molecular size, and an existing quantum algorithm (a boson sampler with displacements) is fraught with experimental difficulties. As an alternative quantum approach, we propose to employ a boson sampler without displacements that offers ease of implementation.” *on Page 7 in SM.*

(2) In **Discussion**, we have added the conclusion about the scalability of the algorithm as

“Our approach enables the generation of molecular vibronic spectra through the modified boson sampling in an optical network. Compared with the traditional boson sampling, our algorithm incorporates an additional weighted summation of the hafnian function. It is noted that the number of summation terms is smaller than the sampling number. As a result, the scalability of

the algorithm mainly depends on the sampling process (see details in **Methods** and **Supplementary Note 2**)." on Page 11.

(3) In **Methods**, we have added some details regarding the sampling process as

"To access the algorithm, it is rewritten as a stochastic sampling problem, considering a probability distribution determined by the boson sampling device. $\mathbf{X}(\mathbf{n}, \mathbf{l})$ is denoted as the frequency of observing the photon number sequence (\mathbf{n}, \mathbf{l}) in N_{smp} samples, which follows the binomial distribution $\mathbf{X}(\mathbf{n}, \mathbf{l}) \sim \mathbf{B}(N_{smp}, Pr(\mathbf{n}, \mathbf{l}))$. In the experiment, the frequency is used to estimate the probability, $Pr(\mathbf{n}, \mathbf{l}) \approx \frac{\mathbf{X}(\mathbf{n}, \mathbf{l})}{N_{smp}}$. Since the events that involve different output photons from each port are mutually exclusive, there is a negative correlation between these events. Then, based on the probability theory [50], the variance of statistical estimation of $\overline{FCF}(\mathbf{n})$ is smaller than the sum of the variance of each term,

$$var(\overline{FCF}(\mathbf{n})) < \mathcal{N} \sum_{\mathbf{l}=0}^{\mathbf{n}} \frac{var\left(\sqrt{\frac{\mathbf{X}(\mathbf{n}, \mathbf{l})}{N_{smp}}}\right)}{\mathbf{l}!}. \quad (4)$$

Based on the central limit theorem and delta method theorem [49],

$$var\left(\sqrt{\frac{\mathbf{X}(\mathbf{n}, \mathbf{l})}{N_{smp}}}\right) \approx \frac{1 - Pr(\mathbf{n}, \mathbf{l})}{4N_{smp}} < \frac{1}{4N_{smp}}. \quad (5)$$

By combining **Eqs. (4)** and **(5)** and considering only those events that happened,

$$var(\overline{FCF}(\mathbf{n})) < \mathcal{N} \sum_{\mathbf{l}=0}^{\mathbf{n}} \frac{var\left(\sqrt{\frac{\mathbf{X}(\mathbf{n}, \mathbf{l})}{N_{smp}}}\right)}{\mathbf{l}!} < \mathcal{N} \frac{t}{4N_{smp}}, \quad (6)$$

where t is the number of summation terms.

If $var(\overline{FCF}(\mathbf{n})) < \epsilon^2$ is required, the number of samples N_{smp} scales as $O(t\epsilon^{-2})$. Meanwhile, t does not increase with the number of vibrational modes in numerical simulations (see details in **Supplementary Note 2**). Therefore, the number of expected samples required to achieve convergence of FCP is restricted by a constant determined by the desired precision."

Comment 25: line 377: non-Condon anharmonic effect -> non-Condon effect.

Answer 25: As suggested by the Reviewer, we have revised the sentence as

"Then, vibronic transitions of naphthalene, phenanthrene, and benzene under the non-Condon region are calculated." on Page 12.

Comment 26: line 362-363: *The sentence is not understandable.*

Answer 26: As suggested by the Reviewer, we will explain this point in details. This inequality applies a theorem from probability theory in two parts.

$$\begin{aligned} \text{var}(\overline{FCF}(\mathbf{n})) &< \text{var}\left(\sqrt{\overline{FCF}(\mathbf{n})}\right), \\ \text{var}\left(\sqrt{\overline{FCF}(\mathbf{n})}\right) &< \mathcal{N} \sum_{\mathbf{l}=0}^{\mathbf{n}} \frac{\text{var}\left(\sqrt{\frac{\mathbf{x}(\mathbf{n},\mathbf{l})}{N_{\text{smp}}}}\right)}{\mathbf{l}!}. \end{aligned}$$

The first part can be explained by the central limit theorem and delta method theorem,

$$\text{var}(\overline{FCF}(\mathbf{n})) \approx 4E\left(\sqrt{\overline{FCF}(\mathbf{n})}\right)^2 \text{var}\left(\sqrt{\overline{FCF}(\mathbf{n})}\right).$$

Since the number of distinct FCFs is the order of $O(\text{poly}(m))$, where m is the mode number, each amplitude $E\left(\sqrt{\overline{FCF}(\mathbf{n})}\right)$ becomes $O(1/\text{poly}(m))$, thus we can presume $\text{var}(\overline{FCF}(\mathbf{n})) < \text{var}\left(\sqrt{\overline{FCF}(\mathbf{n})}\right)$.

The second part can be explained by the properties of covariance. Since events that involve different output photons from each port cannot occur simultaneously, when one event is more likely to happen, the other event is more likely not to happen, thus there is a negative correlation between the random variables. Let's consider a series of k random variables, X_1, X_2, \dots, X_k , which are pairwise negatively correlated. The covariance matrix of these variables is a semi-negative definite matrix, where the diagonal elements represent the variances of the individual variables, and the off-diagonal elements are negative.

When we compute the sum $Y = X_1 + X_2 + \dots + X_k$, the variance of Y can be expressed as

$$\text{var}(Y) = \sum_{i=1}^k \text{var}(X_i) + 2 \sum_{i \neq j} \text{cov}(X_i, X_j).$$

Since the covariance matrix is semi-negative definite matrix, the negative off-diagonal covariances offset the increase in variance. As a result, the variance of Y is typically smaller than the sum of the individual variances of the random variables.

In the revised manuscript, it is discussed as

“Since the events that involve different output photons from each port are mutually exclusive, there is a negative correlation between these events. Then, based on the probability theory [50], the variance of statistical estimation of $\overline{FCF}(\mathbf{n})$ is smaller than the sum of the variance of each term,

$$\text{var}(\overline{FCF}(\mathbf{n})) < \mathcal{N} \sum_{\mathbf{l}=0}^{\mathbf{n}} \frac{\text{var}\left(\sqrt{\frac{\mathbf{x}(\mathbf{n},\mathbf{l})}{N_{\text{smp}}}}\right)}{\mathbf{l}!}. \quad (4)$$

”on Page 13.

Comment 27: Please check variables in italic or in roman. Equations should end with punctuation or comma.

Answer 27: As pointed out by the Reviewer, we have thoroughly revised the manuscript and corrected those equations.

Comment 28: In SM: delta is used for two different purposes.

Answer 28: As pointed out by the Reviewer, one δ has been replaced by \mathbf{K} as the displacement vector on Page 5 in SM.

Comment 29: In SM (Eq. S21-S23): Still, needs more work. It contains errors or typos, B_{ij} . They can be presented better. I think it is not well presented in a logical way. Improve the derivation by focusing on how to tweak the diagonal part out of A and converting the loop hafnian in terms of hafnians so that only the squeezed vacuums are needed in the experiments.

Answer 29: As suggested by the Reviewer, we have corrected these errors and added a detailed description of the derivation from loop hafnian to hafnians in **Supplementary Note 1**. The following figure gives an example of a complete 4-vertex graph with loops to help to understand the relation between $lhaf$ and haf . The 4-vertex graph with loops is expanded to an 8-vertex graph without loops by doubling the number of nodes, and the total number of perfect matchings in the extended graph, which includes 4, 6, and 8 vertices, is the loop hafnian value of the origin graph.

Figure: An exemplary simple case of mapping between loop hafnian and hafnian. The loop hafnian value of a complete 4-vertex graph with loops (\bar{B}) is equivalent to the total number of perfect matchings in the extended graph A , which is 10. The number of perfect matches is three for 4 vertices (the case of the yellow area), six for 6 vertices (the case of the red area), and one for 8 vertices.

We revised the related part in the **Supplementary Note 1** accordingly as

“We define $\mathcal{B} = \begin{bmatrix} 0 & I_n \\ I_m & 0 \end{bmatrix} [I_{2m} - \sigma_Q^{-1}] = \begin{bmatrix} B & 0 \\ 0 & B^* \end{bmatrix}$ with $B = C_L \tanh(\Lambda) C_L^T$, where C_L is the rotation matrix and Λ is the diagonal matrix of squeezing parameter in Doktorov transformation. Then, we obtain $\zeta = \alpha - B\alpha^*$. The P function is the same as **Eq. (S5)**. After substituting the Q function and the P function into **Eq. (S19)** and performing the integration, it yields

$$\begin{aligned}
\text{FCF}(\mathbf{n}) &= \mathcal{N} \prod_{j=1}^m \left(\frac{\partial^2}{\partial \beta_j \partial \beta_j^*} \right)^{n_j} \exp \left[-\frac{1}{2} \boldsymbol{\beta}_v^T \mathcal{B} \boldsymbol{\beta}_v + \boldsymbol{\zeta}_v^T \boldsymbol{\beta}_v \right] \Big|_{\boldsymbol{\beta}_v=0} \\
&= \mathcal{N} \sum_{\mathbf{l}_1=0}^{\mathbf{n}} \sum_{\mathbf{l}_2=0}^{\mathbf{n}} C_{\mathbf{n}}^{\mathbf{l}_1} C_{\mathbf{n}}^{\mathbf{l}_2} \left(\prod_k \zeta_k^{l_{1k}} \zeta_k^{*l_{2k}} \right) \text{haf}(\mathcal{B}^{\mathbf{n}-\mathbf{l}_1, \mathbf{n}-\mathbf{l}_2}) \\
&= \mathcal{N} \left| \sum_{\mathbf{l}=0}^{\mathbf{n}} \left[C_{\mathbf{n}}^{\mathbf{l}} \left(\prod_k \zeta_k^{l_k} \right) \text{haf}(B^{\mathbf{n}-\mathbf{l}}) \right] \right|^2, \tag{S21}
\end{aligned}$$

where \mathcal{N} is a normalized constant and $C_{\mathbf{n}}^{\mathbf{l}}$ denotes the binomial coefficient. Given that \mathcal{B} is formed by duplicating matrix B , the FCF in the second line of **Eq. (S21)** can be expressed as the relation to the hafnian of matrix B , as demonstrated in the final line of **Eq. (S21)**.

According to **Eq. (S21)**, the FCF is represented as a weighted summation of hafnian functions. However, it is not sufficient to establish a bosonic sampler for effectively obtaining the molecular vibronic spectra under this formula. This is because the hafnian value of each submatrix $B^{\mathbf{l}}$ contribute to the FCF(\mathbf{n}) for each $\mathbf{n} > \mathbf{l}$. This implies that even if one has access to the group output of the hafnian values of all submatrices generated by matrix B (which can be efficiently obtained using a bosonic sampler), reconstructing the corresponding molecular vibronic spectra is still difficult due to the need for computing an exponentially large number of spectral lines. Thus, we propose to represent the displacement parameters in terms of extended matrix elements to ensure that each hafnian value of a submatrix only contributes to a single FCF, and theoretically build the connection between FCF in **Eq. (S21)** and the hafnian function of the extended matrix A as

$$\text{FCF}(\mathbf{n}) = \mathcal{N} \left| \sum_{\mathbf{l}=0}^{\mathbf{n}} \frac{1}{\mathbf{l}!} \text{haf}(A^{(\mathbf{n}, \mathbf{l})}) \right|^2, \tag{S22}$$

where A is an expanded matrix that codes the displacement information into generated elements, i.e.,

$$A = \begin{bmatrix} B & Z \\ Z & 0 \end{bmatrix}, \tag{S23}$$

where

$$\begin{aligned}
B &= C_L \tanh(\Lambda) C_L^T, \\
\boldsymbol{\zeta} &= \boldsymbol{\alpha} - B \boldsymbol{\alpha}^*, \\
Z &= \text{diag}(\boldsymbol{\zeta}), \\
\Lambda &= \text{diag}(\boldsymbol{\lambda}), \\
(\mathbf{n}, \mathbf{l}) &= (n_1, n_2, \dots, n_m, l_1, l_2, \dots, l_m). \tag{S24}
\end{aligned}$$

Here, C_L , $\boldsymbol{\lambda}$ and $\boldsymbol{\alpha}$ are corresponded to the Doktorov transformation as $\widehat{U}_{Dok} = \widehat{D}_{\boldsymbol{\alpha}} \widehat{R}_{C_L} \widehat{S}_{\boldsymbol{\lambda}} \widehat{R}_{C_R}$, (\mathbf{n}, \mathbf{l}) is an output pattern, $A^{(\mathbf{n}, \mathbf{l})}$ is a submatrix about (\mathbf{n}, \mathbf{l}) generated by A , and $\boldsymbol{\zeta}$ is the vector formed by $\boldsymbol{\zeta} = \boldsymbol{\alpha} - B \boldsymbol{\alpha}^*$. **Eq. (S22)** will be substantiated in the following section.

The $\text{haf}(A^{(\mathbf{n}, \mathbf{l})})$ is expressed as

$$\begin{aligned}
\text{haf}(A^{(\mathbf{n}, \mathbf{l})}) &= \sum_{M \in \text{PMP}(\mathbf{n}, \mathbf{l})} \prod_{(i, j) \in M} A_{i, j} \\
&= \left(\prod_k \frac{n_k!}{(n_k - l_k)!} A_{k, k+n}^{l_k} \right) \left[\sum_{M \in \text{PMP}(\mathbf{n}-\mathbf{l}, \mathbf{0})} \prod_{(i, j) \in M} A_{i, j} \right]
\end{aligned}$$

$$\begin{aligned}
&= \left(\prod_k \frac{n_k!}{(n_k - l_k)!} \zeta_k^{l_k} \right) \left[\sum_{M \in PMP(\mathbf{n}-1)} \prod_{(i,j) \in M} B_{i,j} \right] \\
&= \frac{\mathbf{n}!}{(\mathbf{n}-1)!} \left(\prod_k \zeta_k^{l_k} \right) \left[\sum_{M \in PMP(\mathbf{n}-1)} \prod_{(i,j) \in M} B_{i,j} \right] \\
&= \frac{\mathbf{n}!}{(\mathbf{n}-1)!} \left(\prod_k \zeta_k^{l_k} \right) haf(B^{\mathbf{n}-1}).
\end{aligned} \tag{S25}$$

Then,

$$\begin{aligned}
\sum_{\mathbf{l}=0}^{\mathbf{n}} \frac{1}{\mathbf{l}!} haf(A^{(\mathbf{n},\mathbf{l})}) &= \sum_{\mathbf{l}=0}^{\mathbf{n}} \left[\frac{\mathbf{n}!}{(\mathbf{n}-\mathbf{l})!\mathbf{l}!} \left(\prod_k \zeta_k^{l_k} \right) haf(B^{\mathbf{n}-1}) \right] \\
&= \sum_{\mathbf{l}=0}^{\mathbf{n}} \left[C_{\mathbf{n}}^{\mathbf{l}} \left(\prod_k \zeta_k^{l_k} \right) haf(B^{\mathbf{n}-1}) \right].
\end{aligned} \tag{S26}$$

Therefore, by substituting **Eq. (S26)** into **Eq. (S21)**, the relation between FCF and hafnian of the extended matrix A via **Eq. (S22)** is obtained. Then, the vibronic spectral profile called the FCP, is obtained by computing the corresponding FCF as

$$FCP(\omega_{vib}) = \sum_n^{\infty} FCF(\mathbf{n}) \delta \left(\omega_{vib} - \sum_k \omega'_k n_k \right). \tag{S27}$$

” on Page 7 in SM.

Comment 30: "combinatorial number"-> "binomial coefficient".

Answer 30: As suggested by the Reviewer, “combinatorial number” has been replaced by "binomial coefficient" as

“where \mathcal{N} is a normalized constant and $C_{\mathbf{n}}^{\mathbf{l}}$ denotes the **binomial coefficient**.” on Page 6 in SM.

Comment 31: Eq. S24: The authors must prove the last equation. The equality works only when the Hafnian is positivesemidefinite. The authors should prove this. Otherwise, the equation is only sometimes acceptable; it would only work for some special cases. Eq. 2 should appear in the main text with an explanation.

Answer 31: As suggested by the Reviewer, the detailed description of the equation derivation is added into the revised manuscript. It can be proven by simulation that the positive-definiteness of the hafnian function can be ignored in the summation process. The detailed analysis about this derivation have been added in the revised manuscript and supplementary.

(1) In **A squeezed vacuum state and linear optical network for approximated molecular vibronic spectra** section, we have added **Eq. (2)** with an explanation as

“Meanwhile, the probability amplitudes of an output pattern are proportional to the square of the hafnian function as $Pr(\mathbf{n}, \mathbf{l}) = \frac{1}{\mathbf{n}!\sqrt{\sigma_Q}} |haf(A^{(\mathbf{n},\mathbf{l})})|^2$, where $\sigma_Q = \sigma + I_{4m}/2$ with I_{4m} being the $4m \times 4m$ identity matrix and σ is the covariance matrix [36, 37]. Moreover, it is observed that the summation in **Eq. (1)** is mainly dominated by one term (see detailed discussion in **Supplementary Note 1**). Therefore, the relationship between $Pr(\mathbf{n}, \mathbf{l})$ and the FCF can be

established, where the sampling results are used to approximately calculate the molecular vibronic spectra. The approximated FCF ($\widetilde{\text{FCF}}(\mathbf{n})$) at 0 K is expressed as

$$\begin{aligned}\widetilde{\text{FCF}}(\mathbf{n}) &= \mathcal{N} \left| \sum_{\mathbf{l}=\mathbf{0}}^{\mathbf{n}} \frac{1}{\mathbf{l}!} |\text{haf}(A^{(\mathbf{n},\mathbf{l})})| \right|^2 \\ &= \mathcal{N}' \left| \sum_{\mathbf{l}=\mathbf{0}}^{\mathbf{n}} \left(\frac{\text{Pr}(\mathbf{n},\mathbf{l})}{\mathbf{l}!} \right)^{\frac{1}{2}} \right|^2,\end{aligned}\quad (2)$$

where \mathcal{N}' is a normalization constant and $\text{Pr}(\mathbf{n},\mathbf{l})$ is the probability of measuring an output pattern (\mathbf{n},\mathbf{l}) . Then, through sampling many FCFs, the approximated FCP at each given vibrational transition frequency (ω_{vib}) is obtained as

$$\widetilde{\text{FCP}}(\omega_{\text{vib}}) = \sum_{\mathbf{n}}^{\infty} \widetilde{\text{FCF}}(\mathbf{n}) \delta\left(\omega_{\text{vib}} - \sum_k \omega'_k n_k\right), \quad (3)$$

where $\{\omega'_k\}$ are the harmonic angular frequencies of the final and initial states and n_k corresponds to the number of photons in the k th mode. Combined with **Eq. (2)**, our algorithm can estimate the FCP by stochastically sampling the known probability distribution for the output modes, and its scaling behavior is described in **Methods** and **Supplementary Note 2.**”
on Page 6.

(2) In **Supplementary Note 1**, we have added detailed derivation as

“The FCF integral is decomposed into a weighted summation of hafnian functions based on the involvement of the displacement operation for each mode in **Eq. (S22)**. It is observed that, for **Eq. (S22)**, only one term significantly contributes to the summation, while the others have negligible contributions in the numerical molecular simulations. Therefore, the FCF is calculated by a simplified equation, whereby only one term with the maximum value is retained in the summation, as shown in **Fig. S1**. The fidelity between the theoretical FCP and the reconstructed profile using the simplified equation is found to be high. This is achieved by randomly generating the transformation matrices U , displacement vector \mathbf{K} , and vibrational frequencies ω and ω' . As shown in the red bar of **Fig. S1b**, it is evident that when \mathbf{K} is small, the dominant term is associated with small \mathbf{l} , whereas, for larger values of \mathbf{K} , the dominant term shifts to large \mathbf{l} . This observation suggests that among various combinations where each mode’s displacement or squeezing operation exerts an influence, only one particular case has a dominant effect on the vibronic transition. It can be explained that, in the summation process, the variation \mathbf{l} adjusts the extent of involvement of the displacement vector \mathbf{K} , and the value of \mathbf{K} significantly influences each term in the summation [**Eq. (S26)**]. Consequently, when \mathbf{l} takes values at a pertinent position correlated with \mathbf{K} , the magnitude of this term becomes substantially greater in comparison to the others. This leads to the summation in **Eq. (S22)** being typically dominated by one term. Therefore, we can disregard the positive-definiteness of the hafnian function and approximate the FCF using absolute values.

Figure S1: **a** Statistical fidelities for different molecular mode sizes. **b** Statistical fidelities and average of selected \mathbf{l} for different displacement regions. Blue bar is the statistical fidelity from the approximation by keeping the term with the maximum value in the summation in Eq. (S22). Red bar is the average of selected terms \mathbf{l} that contribute to the maximum value of the summation in Eq. (S22).

After disregarding the positive definition of the hafnian function, we further deduce that the approximated FCF at 0 K is expressed as

$$\begin{aligned}\widetilde{\text{FCF}}(\mathbf{n}) &= \mathcal{N} \left| \sum_{\mathbf{l}=0}^{\mathbf{n}} \frac{1}{\mathbf{l}!} |\text{haf}(A^{(\mathbf{n},\mathbf{l})})| \right|^2 \\ &= \mathcal{N}' \left| \sum_{\mathbf{l}=0}^{\mathbf{n}} \left(\frac{\text{Pr}(\mathbf{n},\mathbf{l})}{\mathbf{l}!} \right)^{\frac{1}{2}} \right|^2,\end{aligned}\quad (\text{S28})$$

where \mathcal{N}' is a normalization constant, $\text{Pr}(\mathbf{n},\mathbf{l})$ is the probability to measure an output pattern (\mathbf{n},\mathbf{l}) . In the experiment, after computing the Takagi-Autonne decomposition of A , we obtain the unitary \widetilde{U}_C and the squeezing value $\widetilde{\mathbf{r}}$. By encoding these parameters into our boson sampling circuit with the photon number resolving measurement. Finally, the results of the approximate vibronic spectra are obtained as

$$\widetilde{\text{FCP}}(\omega_{vib}) = \sum_{\mathbf{n}} \widetilde{\text{FCF}}(\mathbf{n}) \delta \left(\omega_{vib} - \sum_k \omega'_k m_k \right). \quad (\text{S29})$$

The simulation about the fidelity of the profile using the approximated FCP in Eq. (S29) is depicted in Fig. S2. We randomly generate a set of 14000 virtual molecules, and distribute them across 14 mode sizes. Each mode comprises 1000 molecules. The fidelity of vibronic spectra between theory and our approximation distribution of randomly selected 100 arbitrary virtual molecular spectra per mode size (different colors belong to different dimensions) is displayed in Fig.S2a. It can be seen that the average fidelities (dashed line) do not decrease with the increasing mode size, and the measured average fidelities are always higher than 0.96. We further summarize the statistical accuracies for 14 different mode sizes in Fig. S2b, and verify that the approximation in Eq. (S29) is reasonable.

Fig. S2 **a** Fidelity of vibronic spectra distribution for all random virtual molecules (dots). Dashed lines are the mean values of fidelity per mode size. **b** Summary of statistical fidelity for different molecular mode sizes.

” on Page 7 in SM.

In summary, all comments made by the Reviewer are addressed. The manuscript and supplementary materials have been carefully corrected. We hope our replies and efforts could ensure a more complete and insightful manuscript. We are very grateful for the Reviewer’s constructive comments, which enabled us to greatly improve the manuscript.

Reply to Reviewer 2

We are grateful to the Reviewer for constructive comments and recognition of our experimental achievements and contributions. We appreciate the valuable suggestions, which have catalyzed great improvements to our manuscript. We are happy to address all issues.

Comment 1: *The authors have significantly re-written the manuscript and have addressed many of my concerns. Unfortunately, I believe that there are still some significant omissions to be corrected and clarifications to be added prior to publication. The most glaring omission is the source of Franck-Condon and Herzberg-Teller parameters. These parameters are not inherent properties of a molecule. Rather, they are either derived from electronic structure calculations at a certain level of theory, or they are fit to a molecular spectrum. The former depends on the level of theory employed, and the latter depends on the experimental conditions and resolution. The authors must provide a source for all parameters in Supplementary Note 5. If the authors calculated the parameters themselves, the program and level of theory should be cited. If the parameters come from a previous study, using them without citation is plagiarism.*

Answer 1: As suggested by the Reviewer, the parameters for formic acid and thymine are sourced from Ref. [5], those for naphthalene, phenanthrene, and benzene from Ref. [16], and for Pyrrole from Ref. [17]. All these references have been cited in **Supplementary Note 5** as

“As defined in **Supplementary Note 1**, U represents the Duschinsky matrix, ω and ω' denote the harmonic angular frequencies of the final and initial states, and \mathbf{K} signifies the displacement vector responsible for the molecular structural changes along the normal coordinates. The vibrational frequencies are in cm^{-1} , while the other quantities are dimensionless. The vibronic transition parameters of formic acid and thymine are obtained from Ref. (5), of naphthalene, phenanthrene, and benzene are obtained from Ref. (16) and of pyrrole are obtained from Ref. (17).” on Page 19 in SM.

Comment 2: *The authors have changed their formic acid example to only 4 modes. However, their U matrix is now non-unitary. How did the authors implement a non-unitary transformation in the experiment?*

Answer 2: As pointed out by the reviewer, in this paper, due to some repeated symbols, there may have been some misunderstanding here. The repeated U matrix in the experiment is replaced by symbol V , which is unitary. All the symbols and the repeated ones have thoroughly reviewed to enhance clarity and readability.

The following revisions have been made to the manuscript:

- (1) The U represented the Duschinsky matrix of molecules remains unchanged.
- (2) The U implemented in our experiment is revised to V as

“The squeezing value \mathbf{r} and unitary matrix V are given by the Takagi-Autonne decomposition [38] of matrix A resulting in $A = V^T(\tanh(\mathbf{r}))V$.” on Page 6.

- (3) The U in Doktorov decomposition is revised to C_L as

“Doktorov *et al.* (11) defined a unitary operator (\hat{U}_{Dok}) which performs the Duschinsky transformation as $\hat{a}'^\dagger = \hat{U}_{Dok}^\dagger \hat{a}^\dagger \hat{U}_{Dok}$, where the Doktorov operator can be further decomposed (10) as $\hat{U}_{Dok} = \hat{D}_\alpha \hat{R}_{C_L} \hat{S}_\lambda \hat{R}_{C_R}'$, where $C_L \text{diag}(\mathbf{r}) C_R' = J$ is the singular value decomposition (SVD) of J and $\alpha = \frac{1}{\sqrt{2}} \mathbf{K}$.” on Page 5 in SM.

Comment 3: *In the sentence "... the vibronic spectroscopies of two molecules with a point-group symmetry are successfully simulated experimentally on the quantum photonic chip", I fail to see the relevance of the symmetries. Both molecules belong to the C_s point group, which only has two irreducible representations. The reduction in the number of Franck-Condon active modes is only partially explained by symmetry.*

Answer 3: As pointed out by the reviewer, it is not necessary to mention the symmetry, which has been deleted in the revised manuscript.

Comment 4: *The performance of experiments is very difficult to assess in Fig. 4, especially subfigure c. The authors added that the benzene experiment has a fidelity of 99.4%, but the grey bar for the experiment is only visible on a single peak.*

Answer 4: As pointed out by the reviewer, due to the line style, only one peak can be visible. To enhance clarity in comparing the theoretical and experimental values, we have adjusted the line style in Fig. 4 and included enlarged insets of the smaller peaks. In the revised Fig. 4, it is evident that there is a strong agreement between the theory and experiment for the highest peak, and the error on the smaller peaks is minimal, contributing to a high fidelity. In our experiment, we apply the equation $F = \frac{\sum_i p_i q_i}{\|\mathbf{p}\| \|\mathbf{q}\|}$, where $\{p_i\}$ and $\{q_i\}$ are normalized probability distributions from experiment and theory, yielding fidelities of 99.4%, 98.9%, and 99.4% for naphthalene, phenanthrene, and benzene, respectively. In addition, utilizing the equation $F_2 = \sum_i \sqrt{p_i q_i}$ from Ref. [41], we obtain fidelities of 98.4%, 98.4%, and 93.4% for naphthalene, phenanthrene, and benzene, respectively. Both methods confirm a high fidelity between our experiment and theory. Considering \mathbf{p} and \mathbf{q} are probability distributions, where $\sqrt{p_i}$ and $\sqrt{q_i}$ are related to probability amplitude, the inner product between vectors with $\sqrt{p_i}$ and $\sqrt{q_i}$ from Ref. [41] aligns with the definition of statistical overlap [42] and is more widely accepted equation for fidelity calculation. Thus, we have modified the fidelity calculation equation to $F_2 = \sum_i \sqrt{p_i q_i}$.

We have revised the related part in manuscript as

“The similarity between the experimentally reconstructed and theoretical FCPs is characterized by computing the fidelity of two sequences, $F = \sum_i \sqrt{p_i q_i}$, where $\{p\}$ and $\{q\}$ are the normalized theoretical and experimental probability distributions, respectively [41-43]. A fidelity F of 92.9% is achieved in this case, which is limited by the inevitable flaws in circuit fabrication and operation, photon noise, and photon loss.” on Page 8.

“The measured fidelities with the vibronic spectra of naphthalene, phenanthrene, and benzene are 98.4%, 98.4%, and 93.4%, respectively.” on Page 10.

Figure 4 has also been revised as

Fig. 4 Vibronic spectra with non-Condon effects. Non-Condon and Franck–Condon profiles are obtained from chip distributions programmed according to the vibronic transitions of naphthalene (a, with structure shown in the inset), phenanthrene (b, with the structure shown in the inset), and benzene (c, with the structure shown in the inset). Red bar graphs depict the histogram of experimental energies, whereas blue bars show the theoretical results. Insets are enlarged parts of small peaks.

Comment 5: In the section "Non-Condon effects in molecular spectroscopy", the authors appear to have mixed up references 33 and 34.

Answer 5: As pointed out by the Reviewer, references (33) and (34) have been corrected on Page 15.

In summary, all comments made by the Reviewer are addressed. The manuscript and supplementary materials have been carefully corrected. We hope our replies and efforts could ensure a more complete and insightful manuscript. We are very grateful for the Reviewer's constructive comments, which enabled us to greatly improve the manuscript.

Reply to Reviewer 3

We are grateful to the Reviewer for constructive comments and recognition of our experimental achievements and contributions. We appreciate the valuable suggestions, which have catalyzed great improvements to our manuscript. We are happy to address all issues.

Comment 1: *In this revised paper, the author diligently explained, and corrected all the points raised in the first review. I do not see any major flaws with the experimental setup and the quantum aspects, apart from some imprecise terms that give the impression that the authors sometimes do not have a strong grasp of the subject matter.*

Answer 1: We appreciate the Reviewer's acknowledgment of the significant revisions made to the manuscript to address your concerns. We have made an additional revision to the manuscript, attempting to make it more legible and comprehensive.

Comment 2: *In the supplementary notes, it is more appropriate to use the term "squeezing parameter", or "squeezing value", rather than "squeeze parameter" or "squeeze value".*

Answer 2: As suggested by the Reviewer, the terms have been replaced by "squeezing parameter", or "squeezing value" as

“the calculated squeezing value by second-order correlation measurements can reach around 0.3 (2.6 dB) when input power is around 0.38 mW.” *on Page 14 in SM.*

“In our experiment, the squeezing parameters are set low and losses in the detection channels are significant. Under this approximation, we can provide an estimation equation of the squeezing parameter.” *on Page 16 in SM.*

Comment 3: *Despite this clever idea of doubling the number of squeezed vacuum modes rather than using squeezed coherent states, and the demonstration that this method indeed has a quantum advantage, this advantage seems not as significant compared to the computational resources that the authors need to employ. This seems likely due to the low squeezing parameter of the source and losses.*

Answer 3: As pointed out by the reviewer, we rechecked the calculation of the quantum enhancement and found mistakes leading to this minor quantum advantage. That is, we inadvertently overlooked the loss effects when calculating the fidelity of classical methods. We corrected the error and get $C = 6.8\%$ for formic acid and $C = 7.3\%$ for thymine, which is comparable with values in Ref. [41]. The simulated fidelity difference between the quantum and classical methods versus squeezing values is shown in the following figure, which is consistent with the analysis from Ref. [41].

Figure: The simulated improvement over classical strategies in the fidelity of estimating the FC-profile, using simulated squeezing.

In addition, we can only claim quantum advantage if the performance exceeds that of a classical computer. At this stage of research, we have provided a proof of principle for quantum simulations. However, we have demonstrated the quantum enhancement compared to a classical strategy and could pave a potential way for future applications on molecular simulations.

The following revisions are made in the manuscript as

“We further calculate the difference between the fidelity of the reconstructed FC profile F_Q and the optimal fidelity obtainable from a classical strategy F_C ($C = F_Q - F_C$) to benchmark against the classical simulation [41, 44]. The result shows that an improvement of FCP fidelity to the ideal FCP over the classical strategy ($C = 6.8\%$) is obtained for formic acid molecules.” on Page 9.

“The fidelity is 97.4%, corresponding to $C = 7.3\%$.” on Page 9.

Comment 4: *I do not agree with the first author's argument which explains that the splitting ratio is difficult to set precisely for the squeezed state and the coherent state interference. Indeed, it seems that regarding the programmable interferometer, the difficulty in setting the splitting ratio of the beam splitter of this interferometer should be the same in this case.*

I however agree with the second argument concerning scalability. The phase locking and the coherent state modulation requires indeed more bulk devices.

To summarize, I globally agree with the proposed correction.

Answer 4: We agree with the reviewer’s insights on the implementation of the splitting ratio. However, there still exists a difficulty in preparing squeezed coherent states on a chip: because of the limitation of squeezing level and spectra purity in the integrated photonics devices [24, 26], the interference effects between the coherent and squeezed light are limited and further affect the effects of simulation. Therefore, it is a challenge to create a squeezed coherent state on a chip.

Such discussion is added in the revised manuscript as

“However, in integrated and bulky optical systems, only the vibronic spectra of virtual molecules or actual two-mode molecules have been experimentally realized [24, 25]. In bulky optical systems, the practical implementation of a modulated coherent state and squeezed vacuum state entails bulky components, such as plates, lenses, and modulators, for phase locking, hindering the devices’ scalability [26-28]. For integrated photonic microprocessor chips, the constraints on the level of squeezing and spectral purity of the squeezed source result in limited interference visibility between the coherent and the squeezed light [24, 26]. Consequently, the on-chip realization of vibronic simulations of actual molecules with multiple modes using squeezed coherent states remains an ongoing challenge.” on Page 4.

Comment 5: *The addition of the fidelity of naphthalene, phenanthrene and benzene under the non-Condon regime makes this article more precise and clear.*

Answer 5: We thank the Reviewer for approving this revision.

Comment 6: *The chip shows a small quantum advantage compared to classical resources (1.9% for FC and 1.3 % for non-Condon).*

It would be good if the author can explain why the advantage is low and how it could be improved.

Answer 6: As suggested by the Reviewer, the fidelity of the FC profile depends on the amount of squeezing level, losses, and detection efficiencies. We have added the loss when calculating the fidelity of classical methods and obtained $C = 6.8\%$ for formic acid and $C = 7.3\%$ for thymine, as discussed in **Answer 3**.

The primary factors restricting the current quantum enhancement on our chip are high loss and low squeezing. Concerning the loss, our large-scale, fully programmable optical interferometer network comprises numerous units, contributing to the increased loss. We attempted to address this by optimizing the basic unit design, reducing chip coupling loss, and enhancing detector efficiency. Theoretically, improved fabrication and packaging processes could further minimize loss. Regarding squeezing enhancement, achieving this is challenging in the silicon platform due to the generation of other nonlinear effects impacting photon purity. In the future, switching the chip material to silicon nitride or lithium niobate could improve the quantum enhancement by increasing squeezing levels and possibly reducing loss.

We have added the analysis about strategies for enhancing quantum performance in **Discussion** as

“The reconstructed fidelities of all molecules are higher than 92% with a quantum enhancement, which is limited by the squeezing level, chip losses and detector efficiency. Through waveguide design, increasing etch levels, and using low-loss materials with higher nonlinearity [48, 49], the low losses and high squeezing level in the integrated architecture could further improve quantum enhancements.” on Page 12.

Comment 7: I appreciate the helpful and detailed explanation. Here are typo errors in the supplementary note :

p.10 squeeze value → squeezing value

p.11 the limit of the sum in the normalization condition.

p.11 eq S26 Unclosed brace ‘{‘

Answer 7: As suggested by the Reviewer, all these typos have been corrected as

“the calculated **squeezing value** by second-order correlation measurements can reach around 0.3 (2.6 dB) when input power is around 0.38 mW.” on Page 14 in SM.

“To determine the constant $B(\xi, \eta)$, the normalization condition $\sum_{(n_s, n_i=0)}^{lim} p(n_s, n_i) = 1$ is imposed.” on Page 15 in SM.

“+ $\frac{a_2}{q_2(\xi, \eta)} [b_2 \tanh(\xi)(1 - \eta)]^{2 \max n_s, n_i} + \frac{a_3}{q_3(\xi, \eta)} [b_3 \tanh(\xi)(1 - \eta)]^{2 \max n_s, n_i}$,” on Page 15 in SM.

Comment 8: All the other proposed corrections were implemented. I however see other typo errors:

Supplementary note I :

p.3 There is an open brace $f(g(t)$

$f(g(t) \rightarrow f(g(t))$

p.7 the complicity → the complexity ?

Answer 8: As pointed out by the Reviewer, all these typos have been corrected as

“We then expand the derivatives in Eq. (S6) using Faa di Bruno’s formula, which gives an expansion equation for the n th derivative composition $f(g(t))$ in terms of Bell polynomial $B_{n,k}(x)$ as

$$d^n f(g(t)) = \sum_{k=0}^n (d^k f)(g(t)) B_{n,k}(dg(t), d^2 g(t), \dots).$$

” on Page 3 in SM.

The sentence on Page 7 of SM is deleted in the revised version.

In summary, all comments made by the Reviewer are addressed. The manuscript and supplementary materials have been carefully corrected. We hope our replies and efforts could ensure a more complete and insightful manuscript. We are very grateful for the Reviewer’s constructive comments, which enabled us to greatly improve the manuscript.

REVIEWER COMMENTS

Reviewer #1 (Remarks to the Author):

The authors made some improvements. However, in my opinion, the current manuscript is far from being accepted. I still see strange expressions and ill-cited references (wrong, random, or too broad). I give my last comments to the authors to improve their manuscript. I will not review the manuscript further.

The molecular vibronic problem is considered classically challenging, but it is still being determined whether it is #P-hard in general. It applies to the corresponding loop hafnian. Please make the complexity of the problem clearer when stated. I would refer the authors to a paper by Oh et al. (Nature Physics volume 20, pages225–231 (2024)) .

Make a clear distinction between loop hafnian and hafnian.

Please make sure the Duschinsky rotation is the phenomenon, and the Doktorov operation is the description of it.

Eq.(1) should start with a loop hafnian, then the hafnian expansion follows.

Eq(2): The authors tried to justify the positiveness with numerical tests, but this is not sufficient. If one term is dominant, why don't authors ignore the other negligible terms? Discuss further when the assumption works and when it fails. I understand this work is experimental, and I don't expect a perfect justification, but I want more than the current version. I suggest the authors be advised by a paper by Wang et al. (J. Phys. Chem. Lett. 2022, 13, 6391–6399) for the sign problem.

line 225: Make clearer $F = \sum_i \sqrt{\pi_i q_i}$ part in the main text. How is it directly interpreted as FCF?

Reviewer #2 (Remarks to the Author):

The authors have once again significantly re-written the manuscript. Aside from the experimental data and the general ideas, it reads as an essentially different paper from the original submission. The authors have done no wrong, but I question the discretion and impartiality of the editors in allowing so much revision.

The re-re-written manuscript does an excellent job highlighting and explaining in detail the significant theoretical and experimental contributions in this work. My concerns and comments have been addressed, so at this point I think the manuscript is suitable for publication in Nature Communications with no further changes.

Reviewer #3 (Remarks to the Author):

In this second revision, the authors have implemented the corrections I suggested. While some improvements have been made to enhance the clarity and precision of the research presented, certain concerns regarding the robustness and clarity of the analysis remain.

Response to comments 3 and 6:

The determination of the optimal C factor achievable through the proposed strategy, along with the contributed discussion on how it can be further enhanced, is appreciated. Nevertheless, it would be advisable for the authors to specify that the C factor indicated on page 9 represents the maximum theoretically achievable value at a specific squeezing level (e.g., around 0.3 in this instance), without any losses, or alternatively, indicate the actual C factor measured considering the losses in the setup. This distinction is crucial because the discussion on page 12 might insinuate that the factor at a constant squeezing level could be enhanced by minimizing losses, thereby introducing ambiguity.

Response to comment 4:

I agree with the author's argument . The authors claim that the "spectral purity" of a single-pass squeezed vacuum state decreases the visibility of interferences with a coherent state. Indeed, the broad-spectrum nature of on-chip generated squeezed vacuum states could affect the visibility of interferences with the coherent state, whose spectrum typically mirrors laser properties.

However, I have reservations concerning the term "spectral purity" employed by the author to describe the spectrum of the squeezed vacuum state source. While this term holds significance in optical contexts, it is predominantly associated with lasers or bright

squeezing. Consequently, the sentence: "For integrated photonic microprocessor chips, the constraints on the level of squeezing and spectral purity of the squeezed source result in limited interference visibility between the coherent and the squeezed light [24, 26]." may pose comprehension challenges.

It is suggested that the authors use terms such as: "mode matching" of the two spectrums and "broadband spectrum" for the squeezed vacuum state source to enhance clarity.

In conclusion, the authors have made progress in addressing the corrections and feedback provided, which has helped improve the clarity and precision of their work to some extent. However, the concerns regarding the analysis's ambiguities still persist, casting some doubts on the overall robustness of the research findings. Due to limitations in my expertise, I am unable to provide definitive feedback on other aspects, such as the chemistry discussed in the manuscript.

Manuscript ID: Nature Communications manuscript NCOMMS-23-34789C

Paper title: Large-scale photonic network with squeezed vacuum states for molecular vibronic spectroscopy

Authors: H. H. Zhu, H. S. Chen, T. Chen, Y. Li, S. B. Luo, M. F. Karim, X. S. Luo, F. Gao, Q. Li, H. Cai, L. K. Chin, L. C. Kwek, B. Nordén, X. D. Zhang, and A. Q. Liu

Reply to Reviewer 1

We are grateful to the Reviewer for the constructive comments and recognition of our experimental achievements and contributions. We are happy to address the final comments.

Comment 1: *The authors made some improvements. However, in my opinion, the current manuscript is far from being accepted. I still see strange expressions and ill-cited references (wrong, random, or too broad). I give my last comments to the authors to improve their manuscript. I will not review the manuscript further.*

Answer 1: As pointed out by the Reviewer, we have rechecked the expressions and all ill-cited references have been replaced or removed as follows:

1.1 Ref. [13] has been replaced as

13. Oh C, Lim Y, Wong Y, Quantum-inspired classical algorithms for molecular vibronic spectra. *Nat. Phys.*, **20**, 225–231 (2024).

1.2 The citations, Ref. [49] and Ref. [50], containing incorrect references, have been removed.

1.3 The definitions of some terms have been revised and added as

“The loop hafnian of a particular matrix, which is a quantity related to the perfect matchings of a graph with loops [6, 13, 31].” on Page 5.

“The hafnian (the number of perfect matchings of a graph without loops).” on Page 5.

Comment 2: *The molecular vibronic problem is considered classically challenging, but it is still being determined whether it is #P-hard in general. It applies to the corresponding loop hafnian. Please make the complexity of the problem clearer when stated. I would refer the authors to a paper by Oh et al. (Nature Physics volume 20, pages 225–231 (2024)).*

Answer 2: As pointed out by the Reviewer, Oh et al. (Nature Physics volume 20, pages 225–231 (2024)) proposed a polynomial-time classical algorithm for molecular vibronic spectra. However, in a general scenario where the initial state is a displaced squeezed Fock state, the authors did not find a regime where the approximation error is sufficiently small using their classical algorithm. Considering that classical algorithms only provide partial solutions to the molecular vibronic problem, in the revised manuscript and **Supplementary Note 2**, we clearly discuss the problem's

complexity, elaborate on the current status of classical algorithms, and include the reference mentioned by the Reviewer.

(1) In the **Introduction**, a discussion of the classical algorithm in *Oh et al. (Nature Physics volume 20, pages 225–231 (2024))* has been added as

“A quantum-inspired classical algorithm is also proposed to obtain the partial solution of the molecular vibronic spectra simulation [13].” on Page 3.

(2) In the **Results**, the description of the molecular vibronic problem has been changed to a classically challenging problem and the reference mentioned by Reviewer has been added as

“In classical computing, solving the classically challenging problem of computing the FCF for a given vibronic molecular transition is equivalent to the loop hafnian of a particular matrix, which is a quantity related to the perfect matchings of a graph with loops [6, 13, 31].” on Page 5.

(3) In the **Discussion**, a discussion about the classical and quantum methods for this problem has been added as

“We want to highlight that some classical algorithms can also achieve the same accuracy as compared to quantum approaches in solving the molecular vibronic spectra, such as Gurvits’s algorithm [13, 47, 48], but are limited to specific cases, such as the Fock-state or Gaussian boson sampling with large squeezing. It remains an interesting question whether one can find a classical algorithm to provide a generalized solution in which quantum approaches may offer the solution with potential quantum advantages [13].” Page 12.

(4) In **Supplementary Note 2**, discussions of the complexity of the problem have been revised as

“Then, a boson sampling device is applied to solve the molecular vibronic spectra in quantum chemistry [10]. However, there is a concern that the histogram of boson sampling estimation can also be estimated using several polynomial-time classical algorithms [13 - 15]. Recently, Ref. [16] proposed a quantum-inspired classical algorithm, the generalized Gurvits’s algorithm, and analyzed the applicable and inapplicable conditions of this classical algorithm. The study reveals that the generalized Gurvits’s algorithm can be adapted to estimate the molecular vibronic spectra corresponding to the Fock-state or Gaussian boson sampling with large squeezing [16]. However, no known efficient classical algorithm can approximate the general case whereby the initial state is the Fock state with squeezing and non-zero displacement. As a result, the quantum-vs-classical separation in the computational complexity of the molecular vibronic spectra problem is not completely determined yet. Thus, exploring the molecular vibronic spectra problem using quantum methods is crucial, particularly considering its potential for providing a quantum advantage for molecules that cannot be simulated using classical algorithms.” on Page 10 in SM.

Comment 3: *Make a clear distinction between loop hafnian and hafnian.*

Answer 3: As suggested by the Reviewer, we have explicitly defined loop hafnian and hafnian in the revised manuscript as

“In classical computing, solving the classically challenging problem of computing the FCF for a given vibronic molecular transition is equivalent to the loop hafnian of a particular matrix, which is a quantity related to the perfect matchings of a graph with loops [6, 13, 31].” on Page 5.

“To perform the quantum computation of FCF efficiently in the experiment, we build a relation between the loop hafnian and the hafnian (the number of perfect matchings of a graph without loops).” on Page 5.

Comment 4: *Please make sure the Duschinsky rotation is the phenomenon, and the Doktorov operation is the description of it.*

Answer 4: As suggested by the Reviewer, the definitions of the two terms are confirmed as follows:

- (a) Doktorov operation describes the transformation between the initial and the final vibronic states of a molecule when it undergoes a vibronic transition.
- (b) Duschinsky rotation is the phenomenon, which can be expressed by a linear transformation of creation and annihilation operators, with this transformation being described by the Doktorov operation.

We have explicitly defined the Doktorov operation and the Duschinsky rotation in the revised manuscript as

“For a molecule with n vibrational modes, as shown in **Fig. 1a**, the quantum computation of the vibronic spectra needs to consider the transformation between two potential energy surfaces. The normal coordinates of the final and initial electronic states are linearly related by the Duschinsky transformation [34], which can be represented by the Doktorov operator \hat{U}_{Dok} and further expressed in three quantum operators: squeezing, rotation, and displacement (see detailed in **Supplementary Note 1**).” on Page 5.

Comment 5: *Eq.(1) should start with a loop hafnian, then the hafnian expansion follows.*

Answer 5: As pointed out by the Reviewer, the Eq. (1) has been revised in the revised manuscript as

“The established connection between FCF and the hafnian function of matrix \$B\$ and its extension, matrix \$A\$ , is

$$\begin{aligned} \text{FCF}(\mathbf{n}) &= \mathcal{N} \left| \text{lhaf} \left(\overline{B}^{(\mathbf{n})} \right) \right|^2 \\ &= \mathcal{N} \left| \sum_{\mathbf{l}=0}^{\mathbf{n}} \frac{1}{\mathbf{l}!} \text{haf} \left(A^{(\mathbf{n},\mathbf{l})} \right) \right|^2, \end{aligned} \quad (1)$$

where \overline{B} is a matrix generated by replacing the diagonal entries of B with displacement vector \mathbf{K} , in which B is the matrix correlated to the molecular vibronic parameters (detailed in

Supplementary Note 1). \mathcal{N} is a normalization constant, $lhaf(\cdot)$ is the loop hafnian function of the matrix, $haf(\cdot)$ is the hafnian function of the matrix, and (\mathbf{n}, \mathbf{l}) is an output pattern.” on Page 5.

Comment 6: *Eq(2): The authors tried to justify the positiveness with numerical tests, but this is not sufficient. If one term is dominant, why don't authors ignore the other negligible terms? Discuss further when the assumption works and when it fails. I understand this work is experimental, and I don't expect a perfect justification, but I want more than the current version. I suggest the authors be advised by a paper by Wang et al. (J. Phys. Chem. Lett. 2022, 13, 6391–6399) for the sign problem.*

Answer 6: As pointed out by the Reviewer, when one term is dominant, the other negligible terms can be ignored. However, the dominant term cannot be known in advance, as it is affected by molecular vibronic parameters. Hence, simplifying the summation to one term is not feasible.

As suggested by the Reviewer, we have added two conditions to satisfy the assumption and further discussed a few exceptional molecules that cannot be simulated by our approximation in the revised manuscript and Supplementary Notes.

“Moreover, a sign approximation is used such that the sign of the hafnian in the summation of **Eq. (1)** is ignored (see detailed discussion in **Supplementary Note 1**).” on Page 6.

“The FCF integral is decomposed into a weighted summation of hafnian functions based on the involvement of the displacement operation for each mode in **Eq. (S22)**. It is observed that, under two specific conditions below, it is possible to further refine the FCF approximation by ignoring the positive-definiteness of the hafnian function in **Eq. (S22)** and approximating the FCF using absolute values. These two conditions are: (1) the dominant terms have the same signs in the summation for wavenumbers with large FCF values, exceeding a normalized threshold of 0.2 for the molecules under discussion; (2) conversely, the absolute value of each term in the summation diminishes for wavenumbers with low FCF values, falling below a normalized threshold of 0.05 for the molecules discussed. Fortunately, these conditions are nearly met in most molecular systems, facilitating the implementation of the sign approximation. It can be explained that the resulting FCF value is large when terms in the summation closely align with the molecule's parameters, indicating dominance. As a comparison, for wavenumbers with small FCF values, no single term fully matches the molecule's parameters, resulting in each term's absolute value being small and contributing to the overall diminution of FCF values. To verify this assumption, the FCF is calculated numerically using a simplified equation, whereby only one term with the maximum value is retained in the summation, as shown in **Fig. S1**.” on Page 7 in SM.

“It should be noted that, however, there are also a few exceptional molecules with highly unique structures that defy the approximation in our method. For example, considering a virtual molecule with $B = \begin{bmatrix} 0 & 1 \\ 1 & 0 \end{bmatrix}$, $Z = \begin{bmatrix} 1 & 0 \\ 0 & -1 \end{bmatrix}$, our approximation would yield a high error and low reconstructed fidelity due to its failure to meet the above conditions wherein the absolute value of two terms in the summation possess identical values but opposite signs, consequently leading to a small FCF value. Nonetheless, such molecules with matrices B and Z that adhere to this particular form are exceedingly rare, with none discovered among the 14,000 random molecules

illustrated in **Fig. S2**. This rarity enables us to employ this approximation method for the molecular simulation with effectiveness and generality.” on Page 8 in SM.

In Wang *et al.* (*J. Phys. Chem. Lett.* 2022, 13, 6391–6399), the sign approximation requires the electronic final state to be larger than the electronic initial state of the molecules. This condition is challenging to be met for real molecules. Furthermore, the Duschinsky rotation of molecules requires minimal influence on the sign of the overlap. In the paper's algorithm, the final vibronic states are artificially built without referring to the real molecules. However, our manuscript needs to consider real molecular scenarios. Unlike the reference, our algorithm does not impose such requirements on the molecular parameters, ensuring that FCPs reconstructed for different molecules exhibit high fidelity.

Comment 7: line 225: Make clearer $F = \sum_i \sqrt{p_i q_i}$ part in the main text. How is it directly interpreted as FCF?

Answer 7: As pointed out by the Reviewer, the equation $F = \sum_i \sqrt{p_i q_i}$ used to calculate the fidelity of two FCF sequences has been revised as

“ $F = \sum_i \sqrt{p_i q_i}$, where $\{p\}$ and $\{q\}$ are the normalized theoretical and experimental FCF sequences of the molecules, respectively [42-44].” on Page 9.

We are grateful to the Reviewer for the constructive comments of our paper for publication. We have corrected and polished our manuscript thoroughly to make it more legible and comply with the policies and formatting requirements of Nature Communications.

Reply to Reviewer 2

Comment 1: *The re-re-written manuscript does an excellent job highlighting and explaining in detail the significant theoretical and experimental contributions in this work. My concerns and comments have been addressed, so at this point I think the manuscript is suitable for publication in Nature Communications with no further changes.*

Answer 1: We are grateful to the Reviewer for the recommendation of our paper for publication. We have corrected and polished our manuscript thoroughly to make it comply with the policies and requirements of Nature Communications.

Reply to Reviewer 3

We are grateful to the Reviewer for the constructive comments and recognition of our experimental achievements and contributions. We are happy to address the final comments.

Comment 1: *The determination of the optimal C factor achievable through the proposed strategy, along with the contributed discussion on how it can be further enhanced, is appreciated.*

Nevertheless, it would be advisable for the authors to specify that the C factor indicated on page 9 represents the maximum theoretically achievable value at a specific squeezing level (e.g., around 0.3 in this instance), without any losses, or alternatively, indicate the actual C factor measured considering the losses in the setup. This distinction is crucial because the discussion on page 12 might insinuate that the factor at a constant squeezing level could be enhanced by minimizing losses, thereby introducing ambiguity.

Answer 1: We agree with the reviewer’s insights on specifying the C factor. The C factor is the experimentally measured value considering the losses, rather than the maximum theoretically achievable value without any losses. The related part has been revised and the theoretical value without any losses has been added in the manuscript and **Supplementary Note 7** as

“The result shows an improvement in the fidelity of the experimentally reconstructed FCP to the ideal FCP over the classical strategy ($C = 6.8\%$) for formic acid molecules at the experimental maximum squeezing level (around 0.3). The simulated theoretical fidelity difference between quantum and classical methods versus squeezing values under different loss values for formic acid molecules is shown in Fig. S8, which shows that C is about 10% for the case without considering any losses (at around 0.3 squeezing level).” on Page 9.

As pointed by the Reviewer, we have simulated the fidelity difference between the quantum and classical methods versus squeezing values under different loss values, as shown in Fig. S8. It is noted that quantum enhancement can be improved by minimizing losses.

Fig. S8: Simulated fidelity difference between quantum and classical methods versus squeezing values under different loss values for formic acid molecules.

Comment 2: *I agree with the author's argument. The authors claim that the "spectral purity" of a single-pass squeezed vacuum state decreases the visibility of interferences with a coherent state. Indeed, the broad-spectrum nature of on-chip generated squeezed vacuum states could affect the visibility of interferences with the coherent state, whose spectrum typically mirrors laser properties.*

However, I have reservations concerning the term "spectral purity" employed by the author to describe the spectrum of the squeezed vacuum state source. While this term holds significance in optical contexts, it is predominantly associated with lasers or bright squeezing. Consequently, the sentence: "For integrated photonic microprocessor chips, the constraints on the level of squeezing and spectral purity of the squeezed source result in limited interference visibility between the coherent and the squeezed light [24, 26]." may pose comprehension challenges.

It is suggested that the authors use terms such as: "mode matching" of the two spectrums and "broadband spectrum" for the squeezed vacuum state source to enhance clarity.

Answer 2: As suggested by the Reviewer, the term "spectral purity" has been replaced by "broadband spectrum" in the revised version as

“For integrated photonic microprocessor chips, the constraints on the squeezing level and the broadband spectral characteristic of the squeezed source result in limited interference visibility between the coherent and the squeezed light [24, 26].” on Page 4.

We are grateful to the Reviewer for the constructive comments of our paper for publication. We have corrected and polished our manuscript thoroughly to make it more legible and comply with the policies and requirements of Nature Communications.